# Repairing Reward Functions with Human Feedback to Mitigate Reward Hacking

## Abstract

Human-designed reward functions for reinforcement learning (RL) agents are frequently misaligned with the humans' true, unobservable objectives, and thus act only as proxies. Optimizing for a misspecified proxy reward function often induces reward hacking, resulting in a policy misaligned with the human's true objectives. An alternative is to perform RL from human feedback, which involves learning a reward function from scratch by collecting human preferences over pairs of trajectories. However, building such datasets is costly. To address the limitations of both approaches, we propose Preference-Based Reward Repair (PBRR): an automated iterative framework that repairs a human-specified proxy reward function by learning an additive, transition-dependent correction term from preferences. A manually specified reward function can yield policies that are highly suboptimal under the ground-truth objective, yet corrections on only a few transitions may suffice to recover optimal performance. To identify and correct for those transitions, PBRR uses a targeted exploration strategy and a new preference-learning objective. We prove in tabular domains PBRR has a cumulative regret that matches, up to constants, that of prior preference-based RL methods. In addition, on a suite of reward-hacking benchmarks, PBRR consistently outperforms baselines that learn a reward function from scratch from preferences or modify the proxy reward function using other approaches, requiring substantially fewer preferences to learn high performing policies.

## 1 Introduction

The reward hypothesis states that "all of what we mean by goals and purposes can be well thought of as maximization of the expected value of the cumulative sum of reward" (Sutton & Barto, 2018). This idea underpins much of reinforcement learning (RL): if we can specify the right reward function, then optimizing for it should yield the desired behavior. However, manually designing a reward function that fully captures a human designer's true objectives is rarely possible (Amodei et al., 2016).

One approach is to instead rely on *proxy* reward functions—simpler specifications that reflect the intended but unobservable ground-truth objective. Unfortunately even well-considered proxies, when optimized for by RL, often fail to produce policies that achieve the desired behavior; a failure mode informally known as *reward hacking* (Krakovna et al., 2020; Pan et al., 2022). For example, in an autonomous driving task, maximizing mean velocity—a proxy for traffic flow—could lead vehicles to block highway on-ramps. The common recourse is an iterative, trial-and-error design process. A designer specifies a proxy reward function, trains an agent, observes the resulting behavior, and then manually edits the reward function to remove unwanted incentives (Booth et al., 2023; Knox et al., 2023). We conjecture that while this process can eventually produce usable reward functions and aligned policies, it is slow, ad hoc, and depends on RL expertise that many domain experts do not have. Automating this process would make RL more practical, for example in domains such as pandemic lockdown policy design (Kompella et al., 2020), autonomous driving (Wu et al., 2021; Dosovitskiy et al., 2017), clinical decision making (Man et al., 2014; Petersen et al., 2019; Eastman et al., 2021), energy management (Henry & Ernst, 2021; Orfanoudakis et al., 2024), or tax policy optimization (Mi et al., 2023).

Another path to alignment is to remove the need for any explicit human reward design through learning a reward function from human preferences over trajectories, i.e., reinforcement learning from human feedback (RLHF). However, standard RLHF approaches typically require large datasets

of human preferences, which are often prohibitively costly to collect (Casper et al., 2023). Some work (Novoseller et al., 2020; Pacchiano et al., 2023) has proposed methods for RLHF with cumulative regret guarantees, using strategic exploration under uncertainty to require fewer preferences, but rely on restrictive assumptions such as discrete state-action spaces and specific human preference generation processes. Scaling up uncertainty-based approaches for RLHF to more complex domains is non-trivial and empirical success has been mixed (Ji et al., 2024; Das et al., 2024; Dwaracherla et al., 2024; Mehta et al., 2023).

Human specified reward functions are often misaligned, requiring an informal and manual reward correction process, while RLHF approaches are potentially data intensive. To address these limitations, we introduce **Preference-Based Reward Repair (PBRR)**, an iterative framework for efficiently and automatically repairing a human-specified proxy reward function using preferences. For many tasks, a human can readily specify a proxy reward function that reflects the unobservable ground-truth objective they have in mind, but lacks robustness to the ways in which an RL agent might exploit it. However, a limited number of targeted adjustments to the proxy reward function

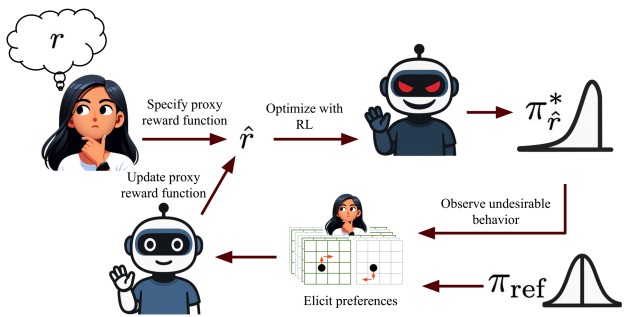

Figure 1: Illustration of Preference-Based Reward Repair (PBRR). A human specifies a proxy reward function $\hat{r}$, which is optimized for with reinforcement learning (RL) to produce a policy $\pi_{\hat{r}}^*$. Preferences between trajectories from $\pi_{\hat{r}}^*$ and a safe reference policy $\pi_{\text{ref}}$ are elicited to identify instances of unaligned behavior. The preferences are used to update $\hat{r}$ and the process repeats, iteratively aligning the proxy reward function with the human's unobservable ground-truth reward function $r$.

may be enough to restore near-optimal behavior. PBRR makes these adjustments as an automated process, in contrast to the manual iterative reward correction process that humans often perform. To automate reward function repair, PBRR leverages two core components: (i) a targeted exploration strategy, which elicits preferences between trajectories generated by the policy trained with the proxy reward function and those from a supplied reference policy, and (ii) a new preference-learning objective to update the proxy reward function only over transitions incorrectly assigned high reward. Figure 1 provides an overview.

On sequential decision process benchmark environments that highlight the challenges of reward hacking (Pan et al., 2022), PBRR significantly outperforms approaches that learn a reward function from scratch from preferences–i.e., standard RLHF—or attempt to repair the proxy reward function using alternative strategies. We also prove that when operating in the tabular settings of past theoretical work, a variant of PBRR matches the cumulative regret bounds of a prior strategic RLHF method (Pacchiano et al., 2023) up to constant terms. Our contributions are three-fold:

- We introduce Preference-Based Reward Repair (PBRR), a method for efficiently repairing a human-specified proxy reward function using a new exploration method and learning objective.
- We prove a variant of PBRR matches, up to constants, the sublinear cumulative regret bounds of Pacchiano et al. (2023) in the same regime.
- We show that PBRR effectively repairs a proxy reward function even when that initial proxy reward induces a substantially suboptimal policy, and consistently outperforms all baselines on a suite of reward-hacking benchmarks.

## 2 BACKGROUND AND SETTING

Consider an MDP $\mathcal{M} \triangleq (S, A, \Omega, \gamma, p_0, r)$ with state space $S$, action space $A$, transition dynamics $\Omega : S \times A \to \Delta(S)$, discount factor $\gamma \in [0, 1]$, and initial state distribution $p_0$. The horizon is $H$ and the ground-truth reward function $r : S \times A \times S \to \mathbb{R}$. $\mathcal{M} \setminus r \triangleq (S, A, \Omega, \gamma, p_0, \_)$ is an environment without a specified reward function. Let $r$ be the (unobservable) ground-truth reward function, $\hat{r}$ a candidate approximation (e.g., learned or specified proxy), and $\tilde{r}$ an arbitrary reward function.

A policy $\pi : S \times A \to [0, 1]$ maps states to action distributions. Its expected discounted return under $\tilde{r}$ from start distribution $p_0$ is $J_{\tilde{r}}(\pi)$. An *optimal policy* for $\tilde{r}$ is any $\pi_{\tilde{r}}^* \in \arg\max_\pi J_{\tilde{r}}(\pi)$. We say $\hat{r}$ is misspecified in environment $\mathcal{M}$ with ground-truth reward $r$ if $J_r(\pi_{\hat{r}}^*) < J_r(\pi_r^*)$. Unless noted, policy

performance refers to expected discounted return under $r$. Let $\tau$ denote a trajectory starting at state $s_0^\tau \sim p_0$: $\tau = (s_0^\tau, a_0^\tau, s_1^\tau, a_1^\tau, ..., s_H^\tau)$. Let a trajectory's return be $\tilde{r}(\tau) = \sum_{t=0}^{H} \gamma^t \tilde{r}(s_t^\tau, a_t^\tau, s_{t+1}^\tau)$ and $\mathcal{T}_\pi$ denote the support of the trajectory distribution induced by $\pi$ from $p_0$.

We learn a reward $\hat{r}$ from trajectory pair preferences. Let

$$\mathcal{D} = \{(\tau_1, \tau_2, \mu)\}_{k=1}^{N}, \quad \mu \in \{0, 1, \tfrac{1}{2}\} \text{ with } 0 : \tau_1 \succ \tau_2, \ 1 : \tau_2 \succ \tau_1, \ \tfrac{1}{2} : \tau_1 \sim \tau_2.$$

As is standard in RLHF (Christiano et al., 2017), we assume a Bradley–Terry preference model:

$$P(\tau_1 \succ \tau_2 \mid \tilde{r}) = \sigma\big(\tilde{r}(\tau_1) - \tilde{r}(\tau_2)\big), \quad \sigma(x) = 1/(1 + e^{-x})$$

Although ubiquitous, this model of noisy rationality may not account for all the ways in which humans fail to act optimally; see Zhi-Xuan et al. (2025) for further discussion.

Unless otherwise stated, we fit $\hat{r}$ by minimizing the cross-entropy loss:

$$\mathcal{L}_{\text{pref}}(\hat{r}; \mathcal{D}_t) = -\sum_{(\tau_1, \tau_2, \mu) \in \mathcal{D}} (1 - \mu) \log P(\tau_1 \succ \tau_2 | \hat{r}) + \mu \log P(\tau_1 \prec \tau_2 | \hat{r}). \tag{1}$$

We assume a preference $\mu$ is elicited over a pair of trajectories, rather than shorter trajectory segments, to mitigate issues relating to misspecified human preference models (see Appendix. A).

## 3 Related Work

Alignment has received extensive attention, particularly in the context of large language models. Here we focus instead on sequential decision processes.

Prior work has explored how to align agent behavior in MDPs despite a human's misspecified reward function under two broad classes of restrictive assumptions. First, some methods assume particular structural properties of the underlying MDP, such as complete knowledge of the human's MDP (Mechergui & Sreedharan, 2024) or that the provided reward function already induces near-optimal behavior (Hadfield-Menell et al., 2017), requiring only additional calibration (Fu et al., 2025). These assumptions do not hold in the environments we study. Second, other methods assume access more demanding human feedback, such as corrective actions (Jiang et al., 2024; Peng et al., 2023), continuous-valued human ratings (Zhang et al., 2024), or feature-attribution–based explanations (Mahmud et al., 2023). Relying on corrective actions would require the human reward designer to provide demonstrations—e.g., controlling a fleet of autonomous vehicles on a highway or determining appropriate pandemic lockdown policies—which demands substantial expertise. Continuous-valued feedback and explanation-based supervision similarly impose a high cognitive burden, limiting who can design aligned reward functions. Our approach, by contrast, requires the human to provide comparative judgments.

Other work infers a posterior over plausible reward functions from human data to mitigate errors in the reward functions (Eisenstein et al., 2023; Mahmud et al., 2023; Coste et al., 2023), and disjoint work shows how such a prior (either learned or provided directly by a stakeholder) can be leveraged for efficient exploration (Novoseller et al., 2020). However it can be challenging for stakeholders to provide Bayesian priors, and learning them may be brittle. We instead only require a stakeholder to provide a single proxy reward function.

In RLHF for large language models, given access to a sufficiently performant reference policy and an estimated but flawed reward function, penalizing KL-divergence between action distributions during training can induce a high-performing policy (see, e.g., Ziegler et al. (2019); Ouyang et al. (2022); Bai et al. (2022); Glaese et al. (2022); OpenAI (2022); Touvron et al. (2023)). In MDPs, Laidlaw et al. (2025) highlight that using different divergences measures can improve policy performance. In contrast, our work focuses on settings where no such high-performing reference policy is available.

Another line of work focuses on efficiently learning a reward function from preferences. Theoretical results for RL in discrete state and action spaces are promising (Novoseller et al., 2020; Pacchiano et al., 2023) but rely on quantifying priors or precise measures of uncertainty over the reward function, which is unclear how to replicate in more complex settings. In large language model settings, preference-based algorithms leveraging coarse approximations of uncertainty or optimism have yielded modest empirical benefit (Mehta et al., 2023; Xie et al., 2024; Das et al., 2024).

Concurrent to our work, Cao et al. (2025) also assume access to a proxy reward function and learn an additive correction term from preferences, demonstrating benefits on robotic manipulation tasks. Our

work differs in several important ways. We study settings where a misspecified proxy reward function induces highly suboptimal behavior. We then propose an exploration strategy that effectively corrects the proxy reward function by leveraging a suboptimal reference policy, as in other RLHF methods. Further, we introduce a new learning objective for repairing a proxy reward function. Together, these components substantially improve performance compared to applying the standard RLHF procedure to update an inputted proxy reward function, which corresponds directly to the baseline of Cao et al. (2025), in the reward-hacking benchmark introduced by Pan et al. (2022). We also provide a theoretical analysis of our approach, whereas Cao et al. (2025) focus solely on empirical results.

# 4 METHODOLOGY

We now present our Preference-Based Reward Repair (PBRR) algorithm. We assume a human stakeholder initially provides a reward function $\hat{r}_{\text{proxy}}(s, a, s')$. PBRR then iteratively aligns this human-specified proxy reward function to their ground-truth objective by eliciting preferences.

Without loss of generality, the ground-truth reward function $r(s, a, s')$ can be written as the proxy reward function $\hat{r}_{\text{proxy}}(s, a, s')$ plus a correction $g(s, a, s')$. Thus, repairing $\hat{r}_{\text{proxy}}$ amounts to learning a transition-dependent correction $g$. At iteration $t$, we elicit a preference batch $\mathcal{D}_t$ and update $g_{t+1}$, yielding the modified proxy reward function:

$$\hat{r}_{t+1}(s, a, s') \triangleq \hat{r}_{\text{proxy}}(s, a, s') + g_t(s, a, s') \tag{2}$$

$\hat{r}_t$ always denotes a modified proxy reward function, while $\hat{r}_{\text{proxy}}$ denotes the original proxy reward function. $g_t(s, a, s')$ is parameterized as a neural network.

This specification offers three benefits. First, the stakeholder need only provide a point-estimate reward function—which will then be corrected—-rather than a full Bayesian prior over reward functions and uncertainties for Bayesian methods. Second, data-efficiency may increase when the complexity of the additive correction term lies in a lower dimensional space than the full reward function. Third, as we will now show, it enables the design of a loss function that explicitly leverages expected properties of the proxy reward function.

In particular, we expect that humans typically provide reward functions that are aligned or overly optimistic.[1] Many cases of reward hacking arise because humans misestimate the cumulative effect of small or multi-objective rewards which can dominate long-term outcomes (e.g., an agent in a racing task learns to loop endlessly through checkpoints to accumulate points, not to finish the race), or because humans mispredict which actions will maximize expected return (e.g., an RL agent exploits a bug in its environment). These types of over-optimistic reward functions cover most examples of reward-hacking from Krakovna et al. (2020). A standard way to learn $g_t(s, a, s')$ would be to minimize cross-entropy (Eq. 1), but this ignores the assumed optimism of the input proxy reward function, potentially increasing corrections for already aligned or optimistic transition rewards and leading to inaccuracies or instability.

**Repairing a proxy reward function** Based on this intuition, we design a loss function for learning the correction term $g$ that regularizes towards only correcting transitions that are incorrectly assigned high reward, conflicting with observed preferences. We first partition the preference dataset into $\mathcal{D}_t^+$, the set that contains all samples where the proxy reward function's induced ranking matches the elicited preference $\mu$, and $\mathcal{D}_t^-$, the set that contains all other samples $(\tau_1, \tau_2, \mu)$:

$$\mathcal{D}_t^+ = \{(\tau_1, \tau_2, \mu) \mid \text{sign}\left(\hat{r}_{\text{proxy}}(\tau_2) - \hat{r}_{\text{proxy}}(\tau_1)\right) = \text{sign}(\mu - 0.5)\} \text{ and } \mathcal{D}_t^- = \mathcal{D}_t \setminus \mathcal{D}_t^+$$

We then learn the corrective term $g$ by minimizing the following three-term loss:

$$\mathcal{L}(g; \hat{r}_{\text{proxy}}, \mathcal{D}_t) \triangleq \mathcal{L}_{\text{pref}}(\hat{r}_{\text{proxy}} + g; \mathcal{D}_t) + \lambda_1 \underbrace{\frac{1}{|\mathcal{D}_t^+|} \sum_{(\tau_1, \tau_2) \in \mathcal{D}_t^+} \left[g(\tau_1)^2 + g(\tau_2)^2\right]}_{\mathcal{L}^+}$$

$$+ \lambda_2 \underbrace{\frac{1}{|\mathcal{D}_t^-|} \sum_{(\tau_1, \tau_2) \in \mathcal{D}_t^-} \left[\mathbb{1}\{\tau_1 \succ \tau_2\} g(\tau_1)^2 + \mathbb{1}\{\tau_2 \succ \tau_1\} g(\tau_2)^2\right]}_{\mathcal{L}^-} \tag{3}$$

---

[1] $\hat{r}$ is overly optimistic if, $\forall (s, a, s'), \hat{r}(s, a, s') \geq r(s, a, s')$

The first term, $\mathcal{L}_{\text{pref}}$, is a standard preference loss that encourages the modified reward function $\hat{r}_{\text{proxy}} + g$ to satisfy the preferences in $\mathcal{D}_t$. The second term $\mathcal{L}^+$ regularizes the correction term $g$ towards zero on trajectory pairs where the proxy reward function agrees with the human preference. This assumes that when the proxy reward function correctly ranks a pair of trajectories, its assigned reward for the transitions in those trajectories is consistent with the ground-truth reward function; $\mathcal{L}^+$ prevents unnecessary adjustments that could degrade an otherwise correct reward signal. Finally, the third term $\mathcal{L}^-$ focuses on trajectory pairs that were misclassified by the modified reward function. Adjusting the correction term $g$ to correctly classify such trajectories is generally underspecified–one could (a) increase the reward for the preferred trajectory, or (b) decrease the reward for the not preferred trajectory. Consistent with our assumptions on the proxy reward function, $\mathcal{L}^-$ prioritizes option (b), regularizing the correction term $g$ to zero on transitions in the preferred trajectories, which will prioritize a negative correction for undesirable behaviors.

To ensure our approach still learns the ground-truth reward function when our assumptions about the specified proxy reward function fail to hold, we decay $\lambda_1$ and $\lambda_2$ over iterations (see Appendix E.6). Although Eq. 3 leverages the assumption that the proxy reward function is optimistic, our algorithm does not require this assumption, nor does our theoretical analysis in Section 5.

---

**Algorithm 1** Preference-Based Reward Repair (PBRR)

---

1: **Input:** Initial proxy reward function $\hat{r}$, reference policy $\pi_{\text{ref}}$, number of iterations $N$
2: Initialize $g_t(s, a, s') \leftarrow 0$ for all $(s, a, s')$
3: **for** $t = 1$ to $T$ **do**
4:     Compute $\pi^*_{\hat{r}_t}$ given the proxy reward function $\hat{r}_t = \hat{r}_{\text{proxy}} + g_t$ [2]
5:     $\pi_1 = \pi^*_{\hat{r}_t}, \pi_2 = \pi_{\text{ref}}$
6:     **if** $C_1 > 0$ **then**
7:         Compute $\Pi_t$, non-dominated policy set
8:         **if** $\pi^*_{\hat{r}_t} \notin \Pi_t$ or $\pi_{\text{ref}} \notin \Pi_t$ or $C_1 f(\pi^*_{\hat{r}_t}, \pi_{\text{ref}}) \leq \max_{\pi_1, \pi_2 \in \Pi_t} f(\pi_1, \pi_2)$ **then**
9:             $\pi_1, \pi_2 = \arg\max_{\pi_1, \pi_2 \in \Pi_t} f(\pi_1, \pi_2)$
10:        **end if**
11:    **end if**
12:    Collect trajectories $\mathcal{T}_{\pi_1}$ and $\mathcal{T}_{\pi_2}$ and sample trajectory pairs $(\tau_1, \tau_2)$ with $\tau_1 \in \mathcal{T}_{\pi_1}, \tau_2 \in \mathcal{T}_{\pi_2}$
13:    Elicit preferences $\mu$ over each pair $(\tau_1, \tau_2)$ and add labeled pairs $(\tau_1, \tau_2, \mu)$ to $\mathcal{D}_t$
14:    Update $\hat{r}_{i+1}$ by learning additive correction $g_{i+1}$ using Equation 3 with $\mathcal{D}_t$
15: **end for**
16: **Output:** Final modified reward function $\hat{r}_T$

---

**Constructing a preference dataset** We next describe how data are gathered to repair the proxy reward function. We assume access to a reference policy, e.g., constructed from heuristics or a previously used policy. Our hypothesis is that such a policy could provide a valuable contrast to when the proxy reward function's induced policy still leads to behavior considered undesirable by the stakeholder.

Accordingly, PBRR prioritizes eliciting preferences between trajectories sampled from the policy that optimizes for the corrected proxy reward function and the reference policy, matching the exploration strategy of Xie et al. (2024). However, in some settings, this exploration strategy may be insufficient to correct the proxy reward function to induce an optimal policy. Therefore, we follow Pacchiano et al. (2023) and define an undominated policy set: the set of policies that remain potentially optimal given the observed data. From this set, we identify the pair of policies with the largest divergence in expected feature values under a weighted covariance norm. If the reference policy and policy optimizing for the corrected proxy reward function have a divergence within a constant of this maximal value, we use the reference policy and corrected proxy reward function's induced policy for exploration. If not, the algorithm can use the maximum divergence policy pair. This strategy enables a principled fallback for additional optimistic exploration when needed, and is analogous to prior work in contextual bandits that defaults to explicit optimistic exploration only when necessary (Bastani et al., 2021). Our PBRR algorithm is detailed in Algorithm 1. The method for constructing the preference dataset, i.e., the exploration strategy, is specified in Lines 5-12.

---

[2]In practice, we train a policy using $\hat{r}_t$ but are not guaranteed to find an optimal policy $\pi^*_{\hat{r}_t}$.

## 5 REGRET ANALYSIS

We now show that if the ground-truth return for a trajectory can be expressed as a linear function of the trajectory embedding, $r(\tau) = \langle \phi(\tau), w^* \rangle$, then PBRR achieves $\sqrt{T}$ cumulative regret. We draw upon recent theoretical results from (Pacchiano et al., 2023). Our key observation is that if the reference policy $\pi_{\text{ref}}$ and optimal policy for the repaired reward function $\pi_{\hat{r}_t}$ lie in the set of possibly optimal policies under the ground-truth reward function given the observed data, and the features induced by their policies maximize the uncertainty with respect to paired feature covariance matrix up to a constant of the maximizing policy pair, then sampling trajectories from the support of $\pi_{\text{ref}}$ and $\pi_{\hat{r}_t}$ will yield bounded cumulative regret within a constant factor of selecting the maximizing uncertainty pair. If these conditions do not hold, our algorithm will instead reduce[3] to selecting the maximizing uncertainty pair. Proofs and additional details are provided in Appendix J.

**Assumption 5.1.** *The trajectory return is linear in a feature trajectory embedding, $r(\tau) = \langle \phi(\tau), w \rangle$.*

**Assumption 5.2.** *Preferences over trajectories are sampled from the Bradley-Terry preference model defined over the difference in return between trajectories.*

**Assumption 5.3.** *We assume that $\|\mathbf{w}^*\| \leq W$ for some known $W > 0$.*

**Assumption 5.4.** *For all trajectories $\tau$ we assume that $\|\phi(\tau)\| \leq B$ for some known $B > 0$.*

In Theorem 5.1 we show that when the dynamics model is known, PBRR inherits the same cumulative regret bounds as Pacchiano et al. (2023) up to constants. We provide full details in Appendix J.

We first define the undominated set of policies as the set of policies that remain potentially optimal under the current uncertainty in the linear reward model parameters:

$$\Pi_t \triangleq \left\{ \pi_i | (\phi(\pi_i) - \phi(\pi))^T w_t + \gamma_t(\delta) \|\phi(\pi^i) - \phi(\pi)\|_{V_t^{-1}} \geq 0 \ \forall \ \pi \right\}$$

We also define the regularized feature covariance matrix with respect to the difference in trajectory feature values: $V_t \triangleq \sum_{l=1}^{t-1} (\phi(\tau_l^1) - \phi(\tau_l^2))(\phi(\tau_l^1) - \phi(\tau_l^2))^T + \kappa \lambda I_d$. Next, we consider the difference in expected feature embeddings of two policies under the known dynamics model, measured in the inverse covariance norm: $f_{mk}(\pi_1, \pi_2) = \|\phi(\pi_1) - \phi(\pi_2)\|_{V_t^{-1}}$. And finally we define a measure of the non-linearity of the sigmoid function over the parameters space: $\kappa \triangleq \sup_{\mathbf{x} \in \mathcal{B}_B(d), \mathbf{w} \in \mathcal{B}_S(d)} \frac{1}{\sigma'(\mathbf{w}^\top \mathbf{x})}$ where $\sigma'$ denotes the derivative and $\mathcal{B}_B(d)$ defines the l2-norm ball of radius $B$ in dimension $d$. We then have:

**Theorem 5.1.** *Let $\delta \leq \frac{1}{e}$ and $\lambda \geq B/\kappa$. Then under Assumptions 5.1, 5.3, 5.2, and 5.4, and that the dynamics model is known, for $f = f_{mk}$ and $\Pi_t = \Pi_{t,mk}$, with probability at least $1 - \delta$, the expected regret of Algorithm 1 is bounded by*

$$Regret_t \leq \tilde{\mathcal{O}}\left( C_1(\kappa\sqrt{\lambda}W + \sqrt{d} + BW\sqrt{d})\sqrt{Td} \right) \tag{4}$$

*where $\tilde{\mathcal{O}}$ hides logarithmic factors in $T, \frac{1}{\delta}, B, \frac{1}{\kappa}, \frac{1}{\lambda}, \frac{1}{d}$.*

We explicitly retain the $C_1$ factor to show that our regret guarantees are slightly looser than the results of Pacchiano et al. (2023), owing to our alternative trajectory selection strategy.

In Theorem 5.2 we show that when the dynamics model is unknown, PBRR also inherits the same cumulative regret bounds as Pacchiano et al. (2023) up to constants. We defer further details to Appendix J, which redefines the undominated policy set to account for uncertainty in the unkown dynamics model $\hat{\mathbb{P}}_t$, and defines $f_u$ to compute the expected trajectory feature difference using $\hat{\mathbb{P}}_t$.

**Theorem 5.2.** *Under Assumptions 5.1, 5.3, 5.2, and 5.4, for $f = f_u$ and $\Pi_t = \Pi_{t,u}$, the regret of Algorithm 1 is bounded by*

$$R_T \leq \tilde{\mathcal{O}}\left( C_1 \left( \kappa d\sqrt{T} + H^{3/2}\sqrt{|\mathcal{S}||\mathcal{A}|dTH} + H|\mathcal{S}|\sqrt{|\mathcal{A}|dTH} \right) \right), \tag{5}$$

*for all $T$ simultaneously with probability at least $1 - 15\delta$, where $\tilde{\mathcal{O}}$ hides logarithmic factors in $\delta, |S|$ and $|A|$.*

---

[3]It is possible to define a variant of our algorithm to handle the various subcases of if $\pi_{\text{ref}}$ is not in the non-dominated policy class $\Pi_t$, if $\pi_{\hat{r}_t}$ is not in the non-dominated policy class $\Pi_t$, or to consider if either either $\pi_{\hat{r}_t}^*$ or $\pi_{\text{ref}}$ can be used as one of the uncertainty-maximizing policy pair; these cases do not impact our theoretical results and so for simplicity we keep the algorithm as is.

Theorems 5.1 and 5.2 suggest that it may often be possible to select the reference policy and policy that optimizes for the current proxy reward function while matching—up to constants—the regret bounds of prior work.

## 6 EXPERIMENTS

We now empirically evaluate PBRR. Our settings largely involve high-dimensional state spaces where the ground-truth reward function is not linear.[4] Defining undominated policy sets for complex, non-linear reward functions learned from preferences is intractable without further restrictions on the policy and reward class, as is finding the best uncertainty-maximizing policy pairs. Therefore, in our empirical results we set $C_1 = 0$, which implies, from Line 6 of Algorithm 1, that for exploration PBRR always uses the reference policy and the policy that optimizes for the corrected proxy reward function. We find that repairing a proxy reward function with PBRR and $C_1 = 0$ induces substantially better performance with fewer preferences than either learning a reward function from scratch via RLHF or repairing a proxy reward function using alternative methods.

### 6.1 ENVIRONMENTS

We evaluate PBRR across the reward hacking benchmark environments used in Pan et al. (2022), summarized in Table 1. More details are in Appendix B. These environments include high-dimensional continuous state and action spaces. For comparison, the state spaces for the Glucose Monitoring, Traffic Control, and Pandemic Mitigation environments are larger than that of the popular Meta-World robotics environments Yu et al. (2020). The action space for Traffic-Control is also larger. All preference labels are sampled from the Boltzmann distribution under the environment's ground-truth reward function over full trajectories. For all environments, the proxy reward function induces substantially sub-optimal performance under the ground-truth reward function.

| Name | Objective | State | Action | H | Ref. policy | $\hat{r}_{\text{proxy}}$ & Summary of $\pi^*_{\hat{r}_{\text{proxy}}}$ | Simulator |
|---|---|---|---|---|---|---|---|
| **Pandemic Mitigation** (Kompella et al., 2020) | Design COVID-19 pandemic lockdown regulations to balance economic and health outcomes | Cont., 312-dim | Disc. $\{-1, 0, 1\}$ | 192 | BC on real-world mitigation strategies from Kompella et al. (2020) | Omits political cost of lockdowns ⇒ policy keeps high lockdown restrictions even at low infection rates | Modified SEIR simulator (Kompella et al., 2020) |
| **Glucose Monitoring** (Man et al., 2014; Fox et al., 2020) | Administer insulin to patient with Type II diabetes to prioritize patient health outcomes | Cont., 96-dim | 1-d Cont. in $[0, 1]$ | 5760 | BC on *few* expert demos (Willis, 1999) | Prioritizes reducing financial cost of treatment ⇒ policy sacrifices patient health for low costs | FDA-approved simulator (Man et al., 2014; Fox et al., 2020) |
| **Traffic Control** (Wu et al., 2021) | Control autonomous vehicle (AV) fleet on highway to maximize traffic flow | Cont., 50-dim | 10-d Cont. in $[0, 1]$ | 300 | BC on *few* human driving demos (Treiber et al., 2000) | Maximizes mean velocity ⇒ AVs block on-ramp so other vehicles can't merge onto highway | FLOW highway simulator (Wu et al., 2021) |
| **AI Safety Gridworld** (Leike et al., 2017) | Water all tomatoes in a grid-world | Disc., 36-dim, | Disc. $\{0, 1, 2, 3\}$ | 100 | Policy trained to only water a subset of tomatoes | Tomatoes look watered when they are not if the agent visits the sprinkler state ⇒ policy never waters all tomatoes | Toy environment from Leike et al. (2017) |

Table 1: Reward-hacking environments, used by Pan et al. (2022). H = horizon. BC = behavior cloning. Cont. = Continuous. Disc. = Discrete.

### 6.2 BASELINES

We compare PBRR against state-of-the-art and natural ablation baselines. More details are in Appendix D. Appendix G.3 presents additional experiments where we update the reference policy across iterations, when relevant.

- **State-Constrained-PPO**   A natural baseline is to optimize for the proxy reward function using the reference policy as a constraint in PPO, with no additional data collection. Laidlaw et al. (2024) showed that using a state-based divergence measure often yields better performance, so we optimize over the set of best-performing divergences amongst their proposals.

---

[4]At least, not in a feature space that is likely to be known to the decision maker in advance.

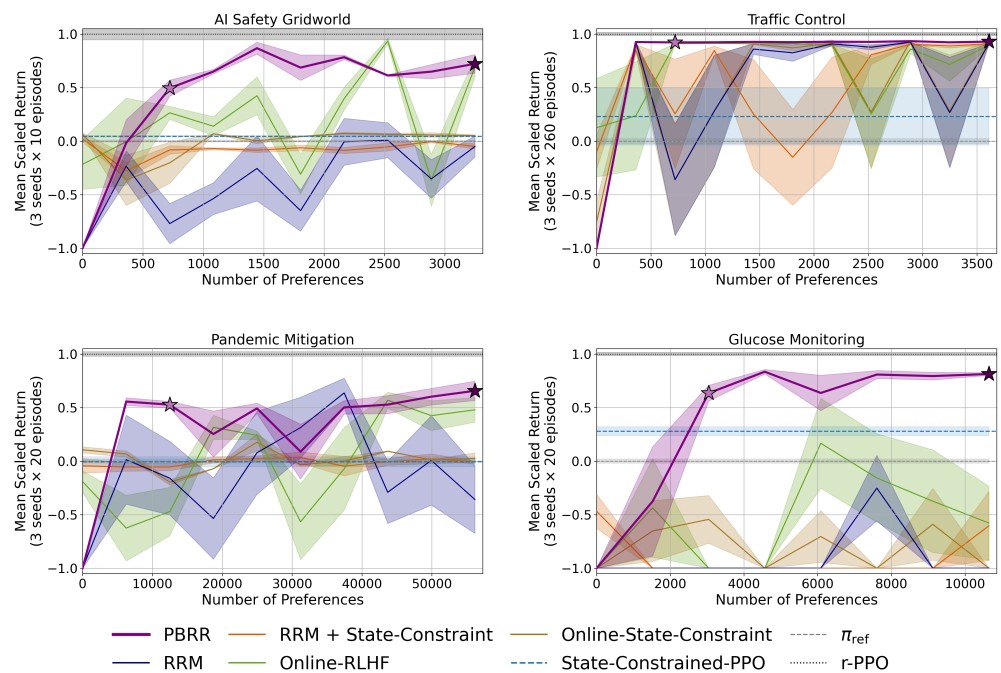

Figure 2: Mean return under the ground-truth reward function achieved by PBRR compared to the baselines from Section 6.2, averaged over 3 random seeds. The mean return for each seed is scaled so that, after scaling, the reference policy has a mean return of 0 and $r$-PPO has a mean return of 1. Values are clipped to $[-1, 1]$, with values below $-1$ indicating very poor performance. Shaded regions indicate the standard error. Unscaled, unclipped returns and further plotting details are provided in Appendix G.7 and Appendix G.1.☆ and ★ mark PBRR's performance after two proxy reward function updates and the final update respectively.

- **Online-State-Constraint**  This method also uses the best divergence-regularized objectives of Laidlaw et al. (2024), but now the reward function is learned from scratch. Preferences are elicited between trajectories sampled from the learned policy and the reference policy.
- **Online-RLHF**  (Christiano et al. (2017)) Their method learns a reward model from scratch and maintains a reward ensemble. Trajectory pairs with the highest predictive uncertainty are selected for preference elicitation. To construct a stronger baseline, the reference policy is used for initial exploration data rather than the initial exploration method of follow-up work (Lee et al., 2021a).
- **Residual Reward Modeling (RRM)**  (Cao et al. (2025)) This method learns the correction term $g$ for Eq. 2 using the standard cross-entropy loss. It elicits preferences over trajectory pairs with the highest predictive uncertainty sampled from the policy optimized for the current proxy reward function. This baseline is a modification of Online-RLHF that leverages the proxy reward function.
- **RRM + State-Constraint**  Here we adapt the Online-State-Constraint baseline to learn the correction term $g$ for Eq. 2 to repair a proxy reward function.
- $r$-**PPO**: PPO using the ground truth reward model. Serves as an oracle upper bound.

### 6.3 MAIN RESULTS: REPAIRING $\hat{r}_{\text{PROXY}}$ WITH PREFERENCES

Results are shown in Figure 2, with additional experiments in Appendix G. Using PBRR to repair the proxy reward function with preferences is more data efficient than learning a reward function *ab initio* without the human-specified prior, and alternative approaches for repairing the proxy reward function. Moreover, PBRR outperforms State-Constrained-PPO, indicating that comparisons to the reference policy enable efficient learning of $g$, even when the reference policy itself is not performant enough to successfully employ any divergence-based method we consider.

★ **Better Jump Start Performance**  After the first two updates of the proxy reward function, PBRR attains significantly higher performance than all baselines in all environments except for Traffic Control, where PBRR matches the performance of Online-RLHF.

★ **Strong Final Performance**  Within the preference budgets considered, PBRR matches or outperforms all baselines in every environment. The performance of RRM indicates that the proxy

reward function alone does not provide enough exploration guidance to learn a correction term even after eliciting a large dataset of preferences; see Appendix H.2 for a qualitative analysis. PBRR overcomes this limitation by leveraging the reference policy to direct exploration.

**Stability**  PBRR is substantially more stable in the AI safety gridworld, Traffic, and Pandemic environments. While other methods can eventually match its performance with enough preferences, their instability makes them impractical without ground-truth evaluation. Unlike PBRR, they show oscillatory or degrading performance as more preferences are collected, since small changes in the reward function can cause large changes in policy performance. Appendix H.1 analyzes this effect, explaining why Online-RLHF is unstable in the AI Safety Gridworld.

**Outperforming the reference policy $\pi_{\mathbf{ref}}$**  Our theoretical analysis of PBRR with $C_1 = 0$ (Appendix K) only guarantees that it asymptotically performs no worse than the reference policy. See Appendix I.1 for illustrative examples. Nevertheless, PBRR empirically induced policies that significantly outperform the reference policy in all environments. The reference policy used by PBRR need not be performant; it must only provide a useful comparison to the proxy reward function's induced policy. In Appendix G.8, we empirically show that a randomly initialized reference policy suffices, with PBRR continuing to match or outperform all baselines when using a randomly initialized reference policy.

**Optimism Assumption**  PBRR leverages the assumption that the proxy reward function is optimistic. However, in the Glucose Monitoring environment, this assumption does not hold; policies that maximize patient health outcomes, i.e., are optimal under the ground-truth reward function, are penalized under the proxy reward function due to their financial cost. Nonetheless, PBRR still outperforms all baselines. Appendix K.7 reports similar findings in the AI Safety Gridworld when repairing a proxy reward function that is not optimistic.

### 6.4 Ablation Study: Preference Learning Objective vs. Exploration Strategy

We also sought to isolate the contributions of our preference-learning objective (Eq. 3) and exploration strategy (Section 4). Figure 3 shows that repairing the proxy reward function with PBRR but using the standard loss in Eq. 1 instead of Eq. 3 yields substantially less stable performance and a lower mean-return under the ground-truth reward function across all environments. Without $\mathcal{L}^+$ and $\mathcal{L}^-$ from Eq. 3, the updated proxy reward function incorrectly assigns a higher reward to the suboptimal actions taken by the reference policy. Appendix H.3 provides a qualitative analysis in the AI safety gridworld, and Appendix G.4 reports further ablations of each regularization term. Conversely, using PBRR's objective (Eq. 3) within the RRM or RRM+State-Constraint baselines fails to match PBRR's data efficiency in all environments except Traffic Control. This shows that both PBRR's preference-learning objective and exploration strategy are important to efficiently repair the proxy reward function with preferences.

## 7 Conclusion

We introduce Preference-Based Reward Repair (PBRR), a framework to repair a human-specified proxy reward function by learning a transition-level correction from preferences. Across a diverse set of benchmarks, spanning autonomous vehicle traffic control, pandemic lockdown regulation design, and insulin regulation for diabetes patients, PBRR achieves higher performance with greater stability and fewer preferences than methods that either learn a reward function from scratch or modify the proxy reward function using alternative strategies. Our ablations show that both components of PBRR are necessary: the exploration strategy that leverages a supplied reference policy to identify transitions over which the proxy reward function is incorrect, and the preference-learning objective that encourages only correcting the proxy reward function on transitions where it incorrectly assigns high reward. Our theoretical analysis further shows that a variant of PBRR attains a sub-linear regret bound comparable with prior work. Additional techniques for improving RLHF data efficiency—such as data augmentation (Park et al., 2022) and intrinsic exploration rewards (Liang et al., 2022)—remain untested in our domains and are orthogonal to our contributions. Future work could explore combining these methods with PBRR to see if further gains in data efficiency are possible. Overall, our results suggest that repairing a human-specified proxy reward function with PBRR, rather than learning a reward function from scratch, is a data-efficient path to alignment in complex, sequential decision making tasks.

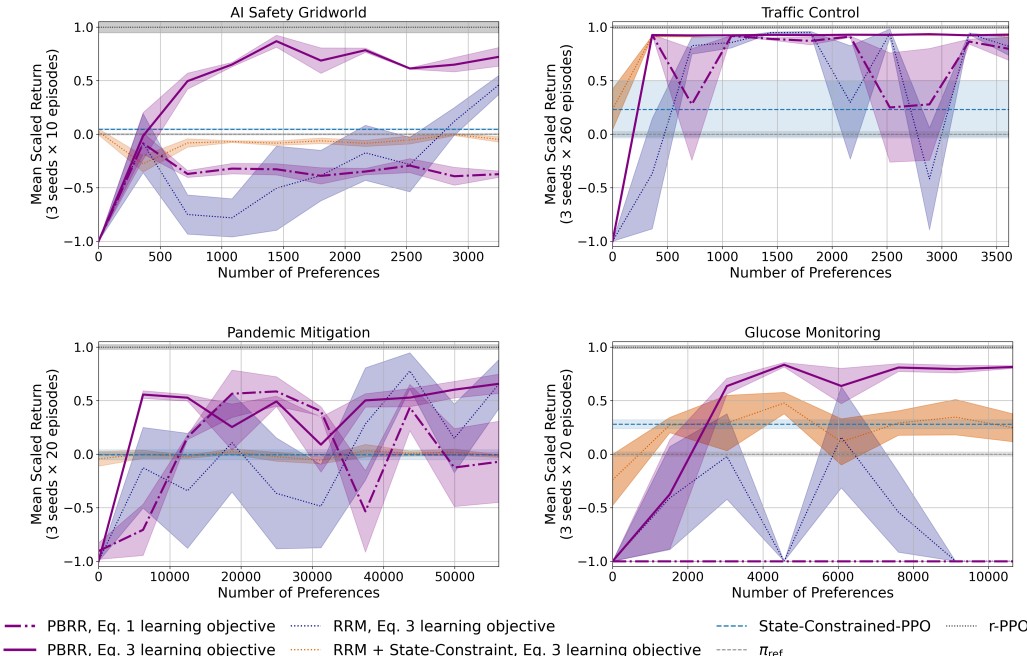

Figure 3: Mean return under the ground-truth reward function achieved by PBRR, compared to (i) PBRR using the standard preference-learning objective (Eq.1) instead of our proposed objective (Eq.3), and (ii) other methods that repair the proxy reward function equipped with our proposed objective. See Figure 2 for plotting details.

**Reproducibility statement** The code to reproduce all experiments is attached with this submission and will be released upon publication.

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

## A  WHY ELICIT PREFERENCES OVER TRAJECTORIES INSTEAD OF TRAJECTORY SEGMENTS?

We focus on eliciting preferences over pairs of trajectories, rather than over pairs of trajectory segments. In practice, eliciting preferences over full trajectories may introduce credit assignment issues that are resolved when eliciting preferences over shorter trajectory segments. The latter is more common than the former (e.g., Christiano et al. (2017)), and we hypothesize that this credit assignment issue results in the noise we observe in our results in Section 6.

Despite the limitation of eliciting preferences over full trajectories rather than shorter segments, this remains a more principled approach for simulating human feedback. In particular, Knox et al. (2022) studied different models of human preferences, finding that the difference in regret between trajectory segments is more predictive of the human preference label than the difference in the sum of rewards. To simulate human preference labels—as we do in this work to avoid collecting preferences from real humans—we therefore should label preferences in accordance with regret under the ground-truth reward function. Unfortunately, for the real-world tasks we consider, computing the regret with respect to the ground-truth reward function is exceedingly difficult and computationally expensive due to the continuous state and action spaces. Consequently, we label preferences over trajectory pairs by the difference in the sum of rewards. These trajectories begin and end in the same state. Therefore, preferences follow the change-in-expected-return model, also proposed by Knox et al. (2022).

Labeling preferences over shorter trajectory segments determined by the difference in sum of rewards— which Knox et al. (2022) call the partial return preference model—- results in preference labels that empirically do not match human judgments. We follow the recommendation of Knox et al. (2022) and avoid using this partial return preference model to simulate preference labels.

The tradeoff however is that credit assignment for reward learning becomes more challenging due to our reliance on eliciting preferences over full trajectories. We argue that simulating higher-fidelity preference labels is more principled, as it better reflects how our methods would perform when no ground-truth reward function is available and real human preferences must be elicited. In other words, trajectory-level preferences allow us to focus on evaluating whether a method can repair a misspecified proxy reward function, rather than conflating this with its capacity to learn a reward function from a misspecified model of human preferences. See Knox et al. (2022) for further discussion on different models of human preferences and their pitfalls, noting that the preference models they consider are equivalent when eliciting preferences over full trajectories that begin and end in the same state in MDPs with deterministic transition dynamics.

## B  ENVIRONMENT DETAILS

We use the same environments and configurations as Laidlaw et al. (2024) and Pan et al. (2022), except for the AI safety gridworld. To make the task from the AI safety gridworld domain harder, we reconfigure the placement of the tomatoes in the grid-world. Full implementation details, including this grid-world configuration, are available in our codebase. We provide a brief description of each environment, as well as its accompanying proxy reward function and reference policy below.

**Pandemic Mitigation**   The agent controls the level of lockdown restrictions imposed on a population by observing COVID-19 test outcomes, as simulated by a modified SEIR model (Kompella et al., 2020). The proxy reward function captures epidemiological and economic outcomes but omits the

political cost associated with aggressive lockdown regulations. Optimizing for the proxy reward function induces a policy that maintains a high lockdown level even when infection rates are low. The supplied reference policy is trained via behavioral cloning on a combination of government strategies used during the pandemic.

Observations are vectors of 312 continuous values; an action is a single discrete value in $\{-1, 0, 1\}$; the horizon is 192 time steps.

**Glucose Monitoring** The agent controls the insulin administered to a simulated patient with Type I diabetes to maintain healthy glucose levels in an FDA approved simulator (Man et al., 2014; Fox et al., 2020). The proxy reward function prioritizes reducing the financial cost of insulin and hospital visits, while the ground-truth reward function is a standard measure of health risk. Optimizing for the proxy reward function induces a policy that minimizes financial costs but not the patient's overall health risk. The supplied reference policy is trained via behavioral cloning on *only a handful* of demonstrations executed by a PID controller tuned by Willis (1999), illustrating a case where only a handful of expert demonstrations are available.

Observations are vectors of 96 continuous values; an action is a single continuous value in $[0, 1]$; the horizon is 5760 time steps.

**Traffic Control** The agent controls a fleet of autonomous vehicles on an on-ramp attempting to merge into traffic on a highway with simulated human drivers (Wu et al., 2021). The proxy reward function prioritizes maximizing the mean velocity of all vehicles. When optimized for, it induces a policy where the autonomous vehicles block the on-ramp, allowing highway traffic to maintain maximum speed while preventing other vehicles from entering. The ground-truth reward function instead prioritizes the mean commute time of all vehicles. The supplied reference policy is trained via behavioral cloning on *only a handful* of demonstrations executed by an Intelligent Driver Model (Treiber et al., 2000) aimed to mimic human driving.

Observations are vectors of 50 continuous values; actions are vectors of 10 continuous values in $[0, 1]$; the horizon is 300 time steps.

**AI Safety Gridworld** In the only toy-task we consider, the agent moves around a grid-world with the objective of watering all tomatoes by visiting tomato-containing states. There exists a sprinkler state that makes the tomatoes look watered, i.e., the agent attains positive reward under the proxy reward function, when the tomatoes are not actually watered, i.e., the agent attains no reward under the ground-truth reward function. This environment was introduced by Leike et al. (2017). The supplied reference policy is trained using the ground-truth reward function in a grid-world layout that contains only a small subset of the tomatoes present in the task we use for evaluation.

Observations are vectors of 36 discrete values; an action is a single discrete value in $\{0, 1, 2, 3\}$; the horizon is 100 time steps.

## C  OPTIMIZING FOR $\hat{r}$ TO AVOID REWARD HACKING

Some prior work (e.g., (Ziegler et al., 2019; Ouyang et al., 2022; Bai et al., 2022; Glaese et al., 2022; OpenAI, 2022; Touvron et al., 2023)) assumes access to a reference policy $\pi_{\mathrm{ref}}$, and then optimizes for the following objective:

$$\text{maximize} \quad J_{\hat{r}}(\pi) \, - \, \beta F(\pi, \pi_{\mathrm{ref}}) \tag{6}$$

where $F$ is some measure of divergence between $\pi$ and $\pi_{\mathrm{ref}}$. For example, $F$ can be defined as the expected KL divergence between the action distributions of $\pi$ and $\pi_{\mathrm{ref}}$:

$$F(\pi, \pi_{\mathrm{ref}}) = (1 - \gamma)\mathbb{E}_\pi\left[\sum_{t=0}^{\infty} \gamma^t \, D_{\mathrm{KL}}\big(\pi(\cdot \mid s_t) \big\| \pi_{\mathrm{ref}}(\cdot \mid s_t)\big)\right]$$

Laidlaw et al. (2024) proposes other divergence measures that dictate how $\pi$ can deviate from $\pi_{\mathrm{ref}}$, which we consider in our empirical results. Eq. 6 applies a divergence penalty between the learned policy $\pi$ and $\pi_{\mathrm{ref}}$ to balance between optimizing for $\hat{r}$ without straying too far from a known behavior distribution, i.e., $\pi_{\mathrm{ref}}$. While this approach can attain better performance under the ground-truth reward function than only optimizing for the misspecified proxy reward function, it

requires a sufficiently performant reference policy or proxy reward function $\hat{r}$ to attain near-optimal performance under the ground-truth reward function. Otherwise, constraining the learned policy to be close to the reference policy as defined by $F$ will fundamentally limit the performance of the resulting policy. On the other hand, reducing the divergence penalty by lowering $\beta$ can weaken the constraint on $\pi$, allowing it to exploit misspecifications in $\hat{r}$ and potentially learn a policy that is substantially sub-optimal with respect to the ground-truth reward function. For this reason, Laidlaw et al. (2024) use a reasonably performant reference policy in their experiments. In contrast, we operate in a setting where no such policy exists; under these conditions, optimizing Eq. 6 with any of the divergence measures considered by Laidlaw et al. (2024) and any choice of $\beta$ is unlikely to yield near-optimal performance with respect to $r$. Therefore, in our setting, $\hat{r}$ itself must be updated.

## D  BASELINES FOR LEARNING AND REPAIRING REWARD FUNCTIONS

Each baseline described below either learns a reward function *ab initio* or assumes access to a manually specified proxy reward function and learns an additive correction term $g(s, a, s')$ as used in Equation 2. In both cases, the additive correction term and the reward function are parametrized as a neural network or an ensemble of neural networks, depending on the baseline. Here we describe how each baseline constructs a batch of trajectory pairs to elicit preferences over. Upon eliciting preferences, the resulting $(\tau_1, \tau_2, \mu)$ samples are added to dataset $\mathcal{D}_t$ and—unless otherwise stated—the reward function or correction term parameters are updated using the standard preference loss in Eq. 1 given $\mathcal{D}_t$.

Our approach and all baselines use Proximal Policy Optimization (PPO) (Schulman et al., 2017) to train a policy with the current estimate of the reward function $\hat{r}$. Like prior RLHF approaches (e.g., Christiano et al. (2017); Lee et al. (2021a;b)), we decouple reward learning and policy learning: PPO samples environment rollouts independently of those used to update the reward function. As a result, differences between methods lie primarily in their reward modeling and data collection strategies, rather than in their policy optimization.

### D.1  ONLINE-RLHF BASELINE

This baseline adapts the methodology of Christiano et al. (2017) and Lee et al. (2021a) to our problem setting. At each iteration, an ensemble of randomly initialized reward models is updated using all collected preferences over trajectories. A new batch of preferences for elicitation is constructed as follows:

At each iteration we sample up to 200 trajectories from the current policy, which are then added to a candidate batch. For a more fair comparison with our approach that assumes access to a safe reference policy, the candidate batch also contains trajectories sampled from the supplied reference policy. We note that the candidate batch initially contains trajectories sampled from the reference policy—-in addition to a policy trained with a randomly initialized reward model—rather than trajectories sampled from a policy pre-trained with the state-entropy objective proposed by Lee et al. (2021a), which is a follow-up work to Christiano et al. (2017); this baseline harnesses the safe reference policy for exploration.

All possible pairs of trajectories are constructed from the candidate batch, and the top $k$ pairs with the highest variance over the predicted preference probabilities—computed via preference model $P(.|\tilde{r})$ with respect to the reward model ensemble—are labeled with preferences. Note that this baseline may elicit preferences between trajectories sampled from the reference policy and the current policy if those pairs have the highest variance in predicted preference probability.

### D.2  RESIDUAL REWARD MODEL (RRM) BASELINE

This baselines mirrors the method of Cao et al. (2025), and matches the Online-RLHF baseline except a correction term $g$ for Eq. 2 is learned instead of learning a reward function *ab initio*. A `tanh` function is also applied to the output of the learned correction term $g$ to match the implementation of Cao et al. (2025). The primary difference between this baseline implementation and Cao et al. (2025) is that they adopt Soft Actor-Critic (SAC) while we use Proximal Policy Optimization (PPO) for a stringent comparison with our other baselines.

### D.3 STATE-CONSTRAINT REWARD LEARNING BASELINES

Here we describe both the Online-State-Constraint and RRM + State-Constraint baselines. These baselines adapt recent insights from alignment research in large language model (LLM) settings, which suggest that constraining the learned policy to remain close to a reference policy can improve performance when learning from preferences. We build on the methodology proposed by Dong et al. (2024), modifying it to suit our setting. In particular, unlike Dong et al. (2024), our policies are not parameterized as LLMs and therefore we cannot exploit stochastic decoding to generate diverse outputs for preference elicitation. To get around this, we sample trajectories from two different policies instead; we sample one trajectory from the learned policy and one from the reference policy.

At each iteration, after learning a reward function from the constructed dataset of preferences, we learn a policy from that reward function using the divergence-regularized objective in Eq. 6. We follow the insights of Laidlaw et al. (2024) and select the divergence measure to be the one that induces the best performance as outlined in Appendix E.4.

We consider two baselines that follow this approach; one that learns a reward function *ab initio* and one that repairs an existing reward function. For each baseline we additionally implement two versions: one that keeps $\pi_{\text{ref}}$ fixed—as presented in Section 6, and one that updates $\pi_{\text{ref}}$ to be the policy learned in the previous iteration—as presented in Appendix E.3.

**Online-State-Constraint**    At each iteration $t$, we roll-out the reference policy $\pi_{\text{ref}}^t$ and the current learned policy $\pi_{const}^t$, where $\pi_{const}^t$ is found by optimizing for the objective in Eq. 6 with $\hat{r}_t$ and $\pi_{\text{ref}}^t$. The specific divergence measure used for this objective is chosen as the measure that induces the best performance as described in Section E.4. $\sqrt{k}$ trajectories are then sampled from $\pi_{\text{ref}}^t$ and $\pi_{const}^t$ respectively, and $k$ exhaustive $(\tau_1, \tau_2)$ pairs are constructed where $\tau_1 \sim \pi_{\text{ref}}^t$ and $\tau_1 \sim \pi_{const}^t$. Preferences are elicited over all $k$ pairs and added to the dataset $\mathcal{D}$. Depending on the baseline variant, the reference policy at the next iteration is moved such that $\pi_{\text{ref}}^{t+1} = \pi_{\text{ref}}^t$ or $\pi_{\text{ref}}^{t+1} = \pi_{const}^t$.

**RRM + State-Constraint**    This baseline follows the same procedure as described above, with the exception that a correction term $g$ is learned and applied to the initial proxy reward function, rather than learning a reward function *ab initio*.

## E    IMPLEMENTATION DETAILS

All experiments were implemented with Python 3.9 and PyTorch 2.7 (Paszke et al., 2019), largely building off of the codebase provided by Laidlaw et al. (2024).

### E.1    POLICY AND REWARD MODEL ARCHITECTURES

We use the same policy network architectures as Laidlaw et al. (2024). For the pandemic, traffic, and AI safety gridworld environments, we use feedforward policy networks with the following architectures: two hidden layers of 128 units for the pandemic environment, four hidden layers of 512 units for the traffic environment, and four hidden layers of 512 units for the AI safety gridworld environment. In the glucose environment, the policy is implemented as a three-layer LSTM, each layer containing 64 units.

When learning a reward function *ab initio*, we parameterize the reward function as a feedforward neural network with 5 hidden layers of 256 units for the pandemic environment and 5 hidden layers of 512 units for the glucose, traffic, and AI safety gridworld environments. For the Online-RLHF baseline, we maintain an ensemble of 5 reward functions. When learning a corrective term $g$ for a specified proxy reward function, we parameterize $g$ as a feedforward neural network with 5 hidden layers of 512 units for the pandemic, glucose, and AI safety gridworld environments, and 3 hidden layers of 256 units for the traffic environment. For the RRM baseline, we maintain an ensemble of 3 reward functions. Table 2 shows the additional hyperparameters used for training the reward function and correction term. The Adam optimizer was used to learn the reward function or correction term parameters with default Pytorch hyperparameter values.

All hyperparameters listed above were found via the following process: for each environment, we constructed a dataset of exhaustive preferences between trajectories sampled from a policy trained with the ground-truth reward function and the proxy reward function respectively, and trajectories

Table 2: Hyperparameters used for learning a reward function *ab initio* and learning a correction term to repair a proxy reward function across all environments.

| Environment | Repair Proxy Reward Function? | Learning Rate | Weight Decay | Epochs |
|---|---|---|---|---|
| Pandemic | True | 0.001 | False | 200 |
| Pandemic | False | 0.0001 | False | 200 |
| Glucose | True | 0.0001 | False | 200 |
| Glucose | False | 0.0001 | True | 200 |
| Traffic | True | 0.0001 | True | 50 |
| Traffic | False | 0.001 | False | 50 |
| AI safety gridworld | True | 0.0001 | True | 200 |
| AI safety gridworld | False | 0.0001 | True | 200 |

sampled from the reference policy. Separately for learning a reward function from scratch and a corrective term for Eq. 2, we performed a grid-search over the following hyperparameter candidates when learning from the constructed dataset of preferences:

- learning rate: [0.001, 0.001]
- weight decay: [True, False]
- number of epochs: [50, 100, 200]
- number of hidden layers: [3, 5]
- number of units per hidden layer: [256, 512]

We chose the hyperparameter values that invoked the lowest loss on a held out test-set. Our objective was to identify the hyperparameters that most effectively enabled the learned reward function to distinguish between trajectories of varying optimality.

### E.2 POLICY AND REWARD MODEL INITIALIZATION

For the glucose, traffic, and pandemic environments, at each iteration, i.e., before any policy is learned, we always initialize the policy weights to be the reference policy weights. Following the methodology of Laidlaw et al. (2024), we randomly initialize the policy weights for the AI safety gridworld environment because otherwise we do not observe reward hacking when optimizing for the initial proxy reward function.

For all environments and experiments, at each iteration we re-initialize the reward function or correction term's weights. We then re-train the reward function or correction term on all collected samples $(\tau_1, \tau_2, \mu) \in \mathcal{D} = \bigcup_t \mathcal{D}_t$, not just the most recently collected batch of preferences $\mathcal{D}_t$.

### E.3 CONTROLLING DIVERGENCE FROM $\pi_{\text{REF}}$

For the State-Constrained-PPO, Online-State-Constraint, and RRM + State-Constraint baselines, we optimize for the objective in Eq. 6 when constructing a policy. To attain the best performance possible given only the proxy reward function and a reference policy, we pick the divergence measure that induces the highest performance under the ground-truth reward function. This methodology, via privileged access to the ground-truth reward function, ensures that the State-Constrained-PPO baseline induces the best performance achievable without updating the proxy reward function. Note that only privileged access to the ground-truth reward function is used to select the divergence measure parameters for the State-Constrained-PPO, Online RLHF + State-Constraint, and RRM + State-Constraint baselines; our approach never uses privileged information.

For the Pandemic environment, we use the same reference policy as Laidlaw et al. (2024), and therefore choose the divergence measure $F$ and constant $\beta$ from Eq. 6 that induces the best performance in their paper: $F$ is the KL-divergence between state occupancy measures and $\beta = 0.06$. For all other environments, we perform a joint search over the choice of occupancy measure—either state or state-action occupancy—and the divergence measure and $\beta$ coefficient specified in Table 3. For each combination, we evaluate performance using the ground-truth reward function and select the

setting that yields the highest mean return across 3 random seeds. For the Traffic environment, $F$ is the KL-divergence between state-action occupancy measures and $\beta = 0.0005$; for the Glucose environment, $F$ is the KL-divergence between state-action occupancy measures and $\beta = 0.06$; for the AI safety gridworld environment, $F$ is the KL-divergence between state-action occupancy measures and $\beta = 0.08$. Note that our search did not include using action-occupancy measures due to their consistently inferior performance in Laidlaw et al. (2024).

Table 3: Coefficient values we searched over for each divergence measure, environment, and occupancy measure.

| Environment | Divergence | Coefficient Values | | | | | | |
|---|---|---|---|---|---|---|---|---|
| | | 1 | 2 | 3 | 4 | 5 | 6 | 7 |
| AI safety gridworld | $\sqrt{\chi^2}$ | 0.0005 | 0.001 | 0.0025 | 0.005 | 0.01 | 0.025 | 0.05 |
| AI safety gridworld | KL | 0.8 | 0.4 | 0.16 | 0.08 | 0.04 | 0.016 | 0.008 |
| Traffic | $\sqrt{\chi^2}$ | 2e-6 | 4e-6 | 1e-5 | 2e-5 | 4e-5 | 1e-4 | 2e-4 |
| Traffic | KL | 0.005 | 0.0025 | 0.001 | 0.0005 | 0.00025 | 0.0001 | 0.00005 |
| Glucose | $\sqrt{\chi^2}$ | 0.0005 | 0.001 | 0.0025 | 0.005 | 0.01 | 0.025 | 0.05 |
| Glucose | KL | 0.0015 | 0.003 | 0.006 | 0.015 | 0.03 | 0.06 | 0.15 |

## E.4 Hyperparameters for PPO

We used the same hyperparameters for PPO as Laidlaw et al. (2024), with two exceptions. To reduce the running time of our approach and all baselines that learn a reward function, which each require training at least one new policy per reward-learning iteration, we reduce the number of PPO training iterations and the PPO batch size. We ensure that with the reduced training batch size and PPO training iterations, optimizing for the proxy reward function still consistently induces reward hacking behavior, and optimizing for the ground-truth reward function still consistently induces the highest achievable expected return or is less than a standard deviation away. The reduced number of iterations and training batch sizes for each environment are shown in Table 4.

Table 4: PPO training hyperparameters for each environment that differ from the ones originally used by Laidlaw et al. (2024).

| Environment | PPO Batch Size | Training Iterations |
|---|---|---|
| Pandemic | 3,840 | 100 |
| Glucose | 10,000 | 150 |
| Traffic | 80,000 | 100 |
| AI safety gridworld | 1,000 | 100 |

## E.5 Learning from preferences experimental details

**How many preferences are elicited per iteration?** For each environment, we elicit $k^2$ preferences per iteration. If a method rolls out two policies $\pi_1$ and $\pi_2$ per iteration, e.g., PBRR samples trajectories from $\pi_{\text{ref}}$ and $\pi_{\hat{r}_t}^*$, then $k$ trajectories are sampled from each policy and preferences are elicited between all $k^2$ pairs $(\tau_1, \tau_2)$ where $\tau_1 \sim \pi_1$ and $\tau_2 \sim \pi_2$. All baselines that roll out a single policy per iteration select $k^2$ trajectory pairs from a candidate batch via ensemble-based uncertainty estimates (see Appendix D for details on how this candidate batch is constructed for each relevant method). $k = 19$ for the AI safety gridworld and traffic environments, $k = 39$ for the glucose environment, and $k = 79$ for the pandemic environment. These values were chosen to balance between distinguishing the sample efficiency of different methods and limiting the computational cost of training a new policy after every reward function update.

**How are trajectories for preference elicitation constructed?** For all environments except glucose, we elicit preferences over full trajectories of length $H$: $H = 100$ for AI safety gridworld, $H = 192$ for pandemic, and $H = 300$ for traffic. In the glucose environment, episodes span $H = 5760$ timesteps. If a simulated patient dies, the episode enters an absorbing state where all observations,

actions, and rewards are zero until the horizon is reached. To avoid out-of-memory issues when learning from preferences, we split each glucose trajectory into three equal segments (each of length 1920), discarding any segments that consist entirely of absorbing-state transitions. Preferences are then elicited over the remaining segments.

**How are preferences labeled?** We label all trajectory pairs with preference labels sampled from the Boltzman distribution given the ground-truth reward function. We use $\gamma = 0.99$ when labeling preferences using $r$, and when learning $\hat{r}$ from preferences. This is the discount factor used by Laidlaw et al. (2024) when learning from $r$ for the same benchmark environments.

### E.6 PBRR $\lambda_1$ AND $\lambda_2$

In Section 4 we outline how the proxy reward function is updated with PBRR via the preference-learning objective in Eq. 3. In practice, we set $\lambda_1 = \lambda_2 = 10$, and then divide both terms by the number of trajectory pairs collected where the proxy reward function's induced ranking agrees with the human preference label, $|\mathcal{D}^+|$. In effect, this decays $\lambda_1$ and $\lambda_2$ as the proxy reward function is repaired. Specifically, at iteration $i$, $\lambda_1^i = \lambda_2^i = \frac{10}{|\mathcal{D}^+|}$.

## F ONLINE-RLHF ABLATIONS

A line of prior work (Lee et al., 2021a; Liang et al., 2022; Metcalf et al., 2024; Park et al., 2022) explored how to improve the data efficiency of RLHF methods across a suite of robotics control tasks. The techniques introduced in these works are complementary to our proposed approach, PBRR, and could in principle be combined with it. We leave such an exploration for future work. In this section, we examine whether the data-efficiency strategies proposed by Lee et al. (2021a); Liang et al. (2022); Metcalf et al. (2024) substantially enhance the initial performance of the Online-RLHF baseline. We find that PBRR continues to outperform the initial performance achieved by Online-RLHF, even when these strategies are applied. We compare to the following methods:

**Online-RLHF** The baseline is used for the main results in Section 6.

**Online-RLHF + Lee et al. (2021a)** Identical to the Online-RLHF baseline, except that, following the unsupervised pre-training procedure introduced by Lee et al. (2021a), we first train an exploration policy using their entropy-maximization objective. Trajectories collected from this exploration policy are then added to the candidate batch used to construct trajectory pairs for preference elicitation. As in Online-RLHF, the candidate batch also includes trajectories sampled from the reference policy.

**Online-RLHF + Liang et al. (2022)** Identical to the Online-RLHF + Lee et al. (2021a) baseline, except that, following the approach of Liang et al. (2022), the learned reward function is augmented with an exploration bonus term that encourages the policy to visit states where the reward estimate is uncertain. This exploration bonus is gradually decayed to 0 once half the total number of preferences used in Figure 2 have been collected.

As shown in Figure 4, PBRR consistently outperforms the initial performance achieved by the Online-RLHF baselines, even when augmented with these data-efficiency techniques, for all environments except Traffic Control. In Traffic Control, PBRR matches the performance of Online-RLHF + Liang et al. (2022) after eliciting a single batch of preferences. We hypothesize that exploration procedures designed to maximize state coverage (e.g., Lee et al. (2021a)) or reward uncertainty (e.g., Liang et al. (2022)) may be less effective in non-ergodic MDPs where most states correspond to undesirable outcomes. For instance, in the Glucose Monitoring environment, many treatment policies yield poor health outcomes for the patient, and exploring such regions of the state space—even if those regions induce the most uncertainty in predicted reward—may provide little information about what constitutes a good treatment policy.

We additionally compare PBRR against the following baseline in the AI safety gridworld:

**Online-RLHF + Metcalf et al. (2024)** Identical to the Online-RLHF + Lee et al. (2021a) baseline, except that, following the approach of Metcalf et al. (2024), a self-supervised dynamics representation is jointly learned with the reward model parameters using their proposed training procedure.

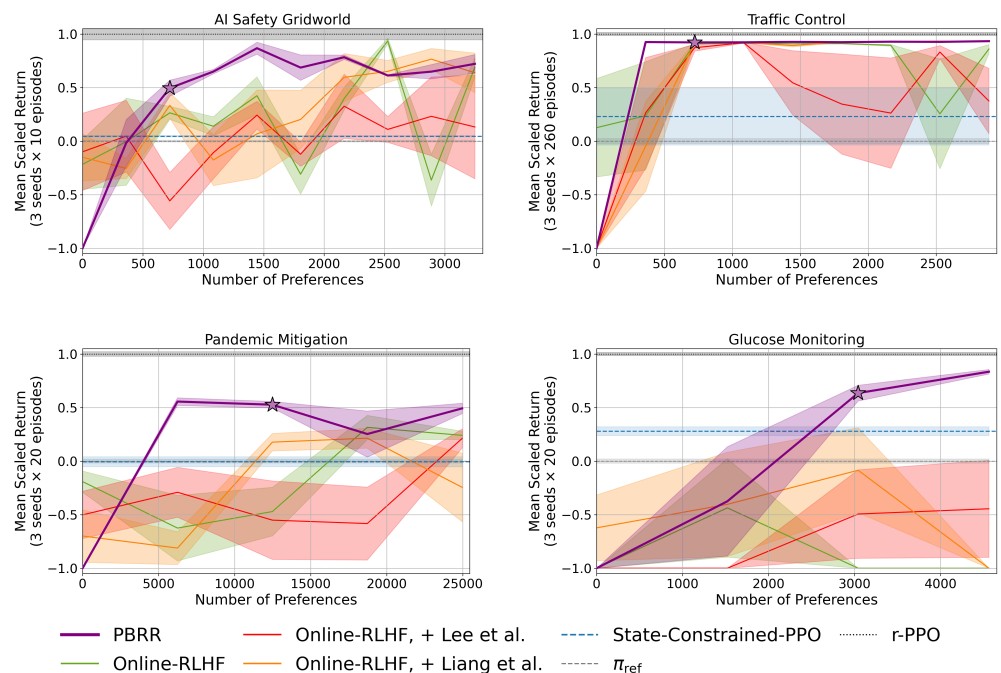

Figure 4: Mean return under the ground-truth reward function achieved by the Online-RLHF baseline and its variants that utilize proposed data-efficiency techniques from Lee et al. (2021a) and Liang et al. (2022). See Figure 2 for plotting details.

The results in Figure 5 show that Online-RLHF + Metcalf et al. (2024) underperforms PBRR and Online-RLHF; we suspect that the learned dynamics aware representation is not sufficient to learn a reward function over.

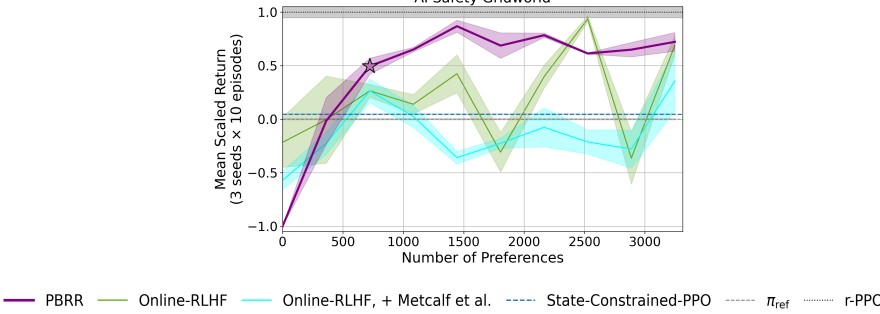

Figure 5: Mean return under the ground-truth reward function achieved by the Online-RLHF baseline and the Online-RLHF + Metcalf et al. (2024). See Figure 2 for plotting details.

## G  PLOTTING DETAILS AND ADITIONAL RESULTS

### G.1  PLOTTING DETAILS

For Figures 2 and 3, for each seed we roll out a policy for the number of episodes specified in Appendix Table 5. We compute the mean return over all episodes with respect to the ground-truth reward function. We then scale the mean return for policy $\pi$ as follows:

$$\hat{J}_{r\text{-scaled}}(\pi) = \frac{\hat{J}(\pi) - \hat{J}_r(\pi_{\text{ref}})}{\hat{J}_r(\pi_r^*) - \hat{J}_r(\pi_{\text{ref}})} :$$

where $\hat{J}_r$ is the empirical mean return under the ground-truth reward function $r$. We clip $\hat{J}_{r\text{-scaled}}(\pi)$ to be in range $[-1, 1]$. We perform this clipping for visual clarity, noting that any policy with a scaled return less than 0 does worse than the supplied reference policy, and any policy with a scaled return less than $-1$ is considerably sub-optimal. In Figures 2 and 3, we plot the resulting mean scaled return over 3 seeds.

| Environment | Number of Episodes |
|---|---|
| Traffic | 260 |
| Pandemic | 20 |
| Glucose | 20 |
| AI safety gridworld | 10 |

Table 5: Number of episodes used to compute the mean return.

### G.2 RETRAINING A POLICY WITH AN UPDATED PROXY REWARD FUNCTION

McKinney et al. (2023) show that reward functions learned online from human feedback can fail to re-train new policies initialized from scratch with a different random seed. To test whether PBRR's updated proxy reward function exhibits similar fragility, we reinitialize new policies with different seeds and train the policies using the repaired reward functions learned in Section 6. We also extend PPO training beyond the number of PPO steps used when learning the additive correction term with PBRR, probing whether the updated proxy reward function could induce undesirable behavior if optimized for longer. Figure 6 presents these results: in the Traffic Control, Glucose Monitoring, and Pandemic environments, retraining with the repaired proxy reward function still yield policies that induce near-optimal performance. In the Pandemic Mitigation environment, training PPO for more steps even yields an improvement. In the Glucose Monitoring environment, however, the performance of the policy induced by the repaired proxy reward function deteriorates when trained with substantially more RL steps than were used during reward learning. These findings broadly suggest that PBRR's repaired proxy reward function is robust to policy reinitialization and to longer RL optimization, but the number of RL steps used while updating the proxy reward function can play a critical role in determining whether the learned signal remains valid under extended training. In other words, when re-training a new policy with a repaired proxy reward function, it is likely best to train that policy with the same number of RL steps as used when repairing the proxy reward function.

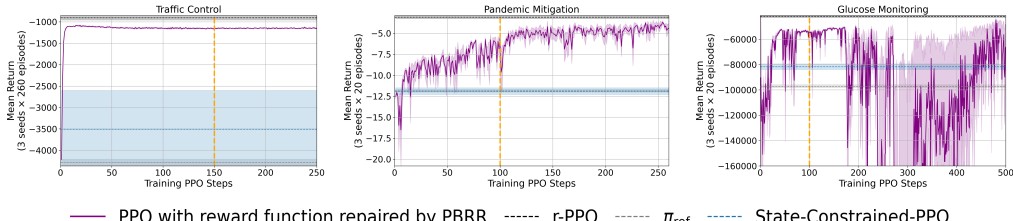

Figure 6: Mean return under the ground-truth reward function achieved by re-training a newly initialized policy with the repaired proxy reward function learned by PBRR. The vertical line marks the number of steps used to train the policies when updating the proxy reward function in Algorithm 1; performance beyond this point illustrates the robustness of that updated proxy reward function to extended optimization. Results are averaged over trajectories sampled from policies trained with the updated proxy reward function across 3 random seeds.

### G.3 COMPARISONS AGAINST BASELINES THAT UPDATE $\pi_{\text{REF}}$

Figure 2 compares PBRR against the Online-State-Constraint and RRM + State-Constraint baselines respectively, which each constrain the learned policy to $\pi_{\text{ref}}$. Section D details these baseline implementations. Here we compare PBRR against those baselines, except we update $\pi_{\text{ref}}$ every iteration to be the policy constructed at the previous iteration as outlined in Section D. We refer to these additional baselines as Online-RLHF + Moving-State-Constraint and RRM + Moving-State-Constraint.

Our goal with these additional baselines is to attempt to overcome a fundamental limitation of the Online-State-Constraint and RRM + State-Constraint baselines, namely that they may fundamentally

limit the performance of the induced policy by penalizing divergences from a fixed $\pi_{\text{ref}}$. Figure 7 below shows that updating $\pi_{\text{ref}}$ as described in Section D does not lead to a substantial improvement in performance; PBRR still remains the most performant approach in all environments.

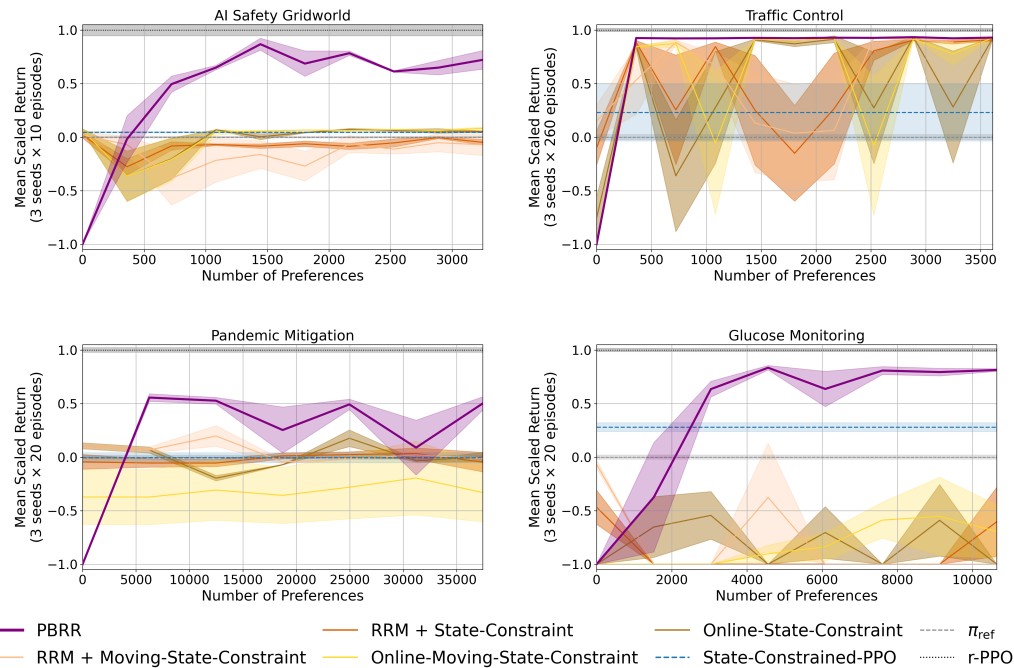

Figure 7: Mean return under the ground-truth reward function achieved by the Online-State-Constraint and RRM + State-Constraint baselines, and variants of those baselines that update $\pi_{\text{ref}}$ used for the state-constraint at each iteration as described in Section D. See Figure 2 for plotting details.

### G.4 ABLATING PBRR'S REGULARIZATION TERMS

PBRR's preference learning objective in Eq. 3 consists of two regularization terms, $\mathcal{L}^-$ and $\mathcal{L}^+$. In Figure 3 we empirically show that, without these regularization terms, PBRR performs relatively poorly in both stability and achieved mean return. Here we investigate the individual impact of $\mathcal{L}^-$ and $\mathcal{L}^+$ respectively. Figure 8 illustrates that PBRR with both regularization terms matches or outperforms all alternatives across environments. The relative effect of using only $\mathcal{L}^-$ or $\mathcal{L}^+$ is environment dependent. PBRR's performance is only degraded by removing a regularization term in the Pandemic Control and Glucose Monitoring environment: removing $\mathcal{L}^+$ causes the larger drop in Pandemic, while removing $\mathcal{L}^-$ has the greater effect in Glucose.

### G.5 PBRR WITH INTRA-POLICY PREFERENCES

PBRR, as implemented for the results in Section 6, elicits preferences over trajectory pairs where one trajectory is sampled from $\pi_{\hat{r}_t}$ and the other from $\pi_{\text{ref}}$. Here we investigate adding intra-policy preferences, i.e., preferences over trajectory pairs where both trajectories are sampled from $\pi_{\hat{r}_t}$. Figure 9 plots the results when half of the elicited batch is intra-policy preferences. We find that including these preferences degrades data-efficiency and stability. In particular, the performance in the early iterations of the PBRR + intra-policy preferences approach is worse than PBRR in the AI Safety Gridworld and Pandemic environment, and performance decreases in later iterations in the Glucose environment. For the Pandemic environment, PBRR + intra-policy preferences eventually outperforms PBRR, suggesting a potential benefit to including intra-policy preferences. Future work should explore how to leverage this additional data to improve PBRR's performance without reducing data-efficiency or learning stability in environments where PBRR already performs well.

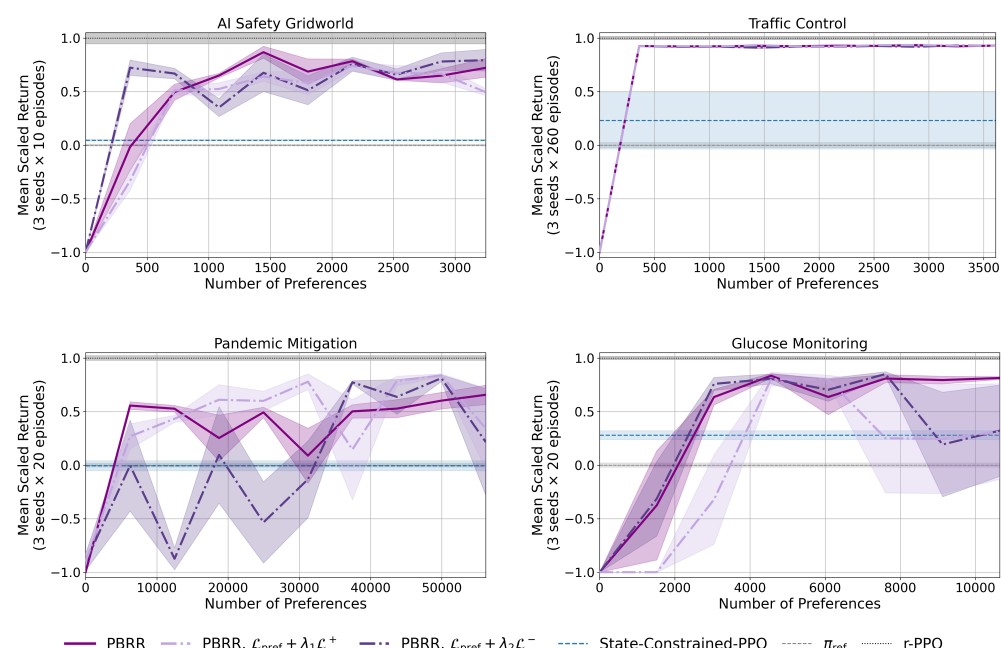

Figure 8: Mean return under the ground-truth reward function achieved by PBRR, compared to PBRR using the standard preference-learning objective $\mathcal{L}_{\text{pref}}$ (Eq.1) with (i) only the $\mathcal{L}^-$ regularization term, and (ii) only the $\mathcal{L}^+$ regularization term. Note that PBRR uses the preference learning objective in Eq. 3 with all three terms: $\mathcal{L}_{\text{pref}} + \lambda_1 \mathcal{L}^+ + \lambda_2 \mathcal{L}^-$. See Figure 2 for plotting details.

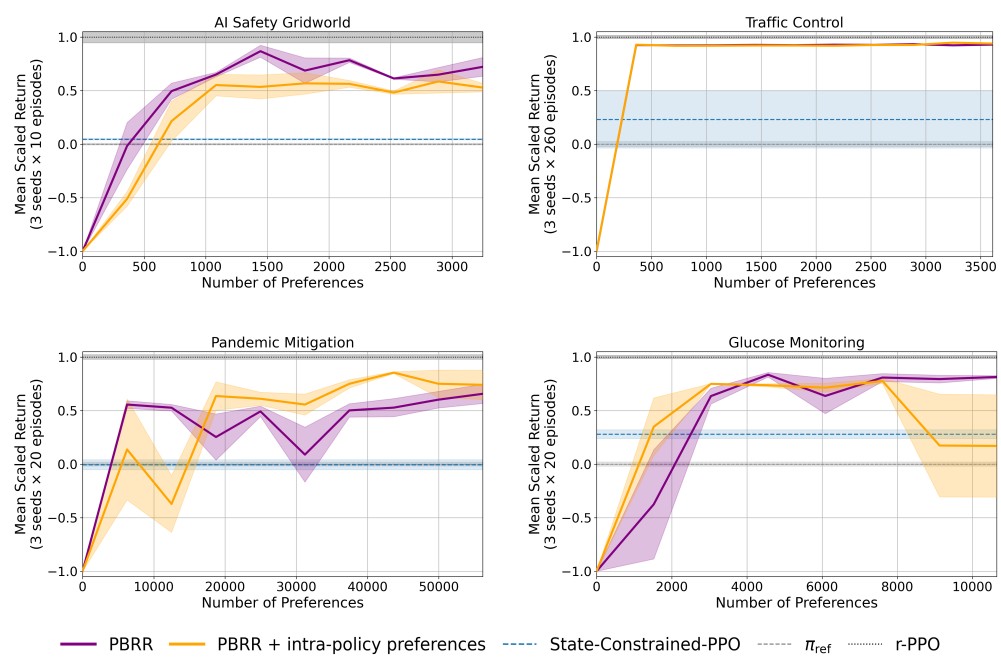

Figure 9: Mean return under the ground-truth reward function achieved by PBRR, compared to PBRR + intra-policy preferences. See Figure 2 for plotting details.

### G.6 PBRR WITH A PESSIMISTIC PROXY REWARD FUNCTION

In Section 4 we discuss a key assumption behind PBRR's preference learning objective: the human-specified proxy reward function is either aligned or optimistic. Here we investigate PBRR's performance in the AI safety gridworld when repairing a pessimistic proxy reward function. In particular, we construct a new proxy reward function that is the same as the proxy reward function used for the AI safety gridworld in Section 6, except that it assigns a negative reward for visiting $\frac{3}{9}$ tomato-containing states—which the ground-truth reward function assigns a positive reward for visiting. As such, this proxy reward function breaks our optimism assumption. We plot the results in Figure 10. PBRR eventually attains near-optimal performance but requires more preferences than when repairing an optimistic proxy reward function, as assumed in Figure 2. We note that even when the optimism assumption does not hold, PBRR still outperforms other methods for repairing the proxy reward function.

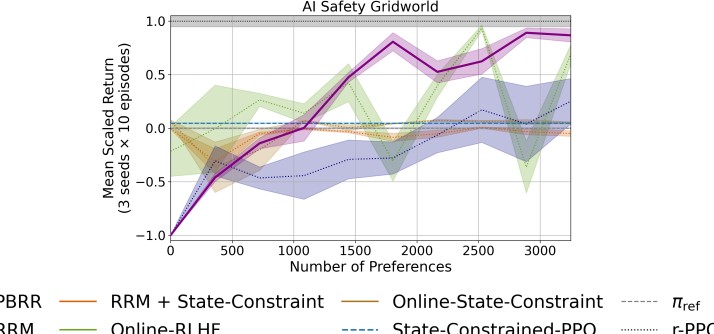

Figure 10: Mean return under the ground-truth reward function achieved by PBRR compared to the baselines from Section 6.2. The proxy reward function here does not follow the optimism assumption leveraged by PBRR. See Figure 2 for plotting details.

### G.7 UNSCALED MAIN RESULTS

In Figures 2 and 3 we scale and clip the plotted mean return, as detailed in Appendix G.1, for clearer visual comparison. In Figure 11 we plot the unclipped, unscaled mean-return for PBRR and the baselines that learn a reward function *ab initio*—Online-RLHF and Online-State-Constraint. In Figure 12 we plot those results for PBRR and the baselines that learn to repair a proxy reward function—RRM and RRM + State-Constraint. In Figure 13 we plot those results for the ablations from Figure 3.

### G.8 PBRR WITH A RANDOMLY INITIALIZED REFERENCE POLICY

PBRR requires a reference policy for preference elicitation. For all prior experiments, unless otherwise stated, we use a realistic reference policy that a human could provide such as one trained with a handful of human demonstrations or from imperfect hand-written rules. Here, we investigate PBRR's performance when using a randomly initialize reference policy instead. Figure 14 shows that performance is largely unchanged; the quality of the reference policy used by PBRR is not judged by its performance, but rather its coverage of the state-action space relative to the policy induced by the proxy reward function. These results imply that for many tasks, a sufficient reference policy should always be available by simply using a randomly initialized policy.

### G.9 EVALUATING PBRR OVER ADDITIONAL RANDOM SEEDS

We aim to establish 95%-confidence intervals for the main results in Figure 2. As a step towards this, we evaluate PBRR across 10 seeds in the Pandemic Mitigation environment. We compare PBRR against the two best performing baselines—Online-RLHF and RRM. Due to computational constraints, we restrict this analysis to the first two updates of the proxy reward function. As shown in Table 6, after the first update PBRR produces a reward function that yields statistically significantly

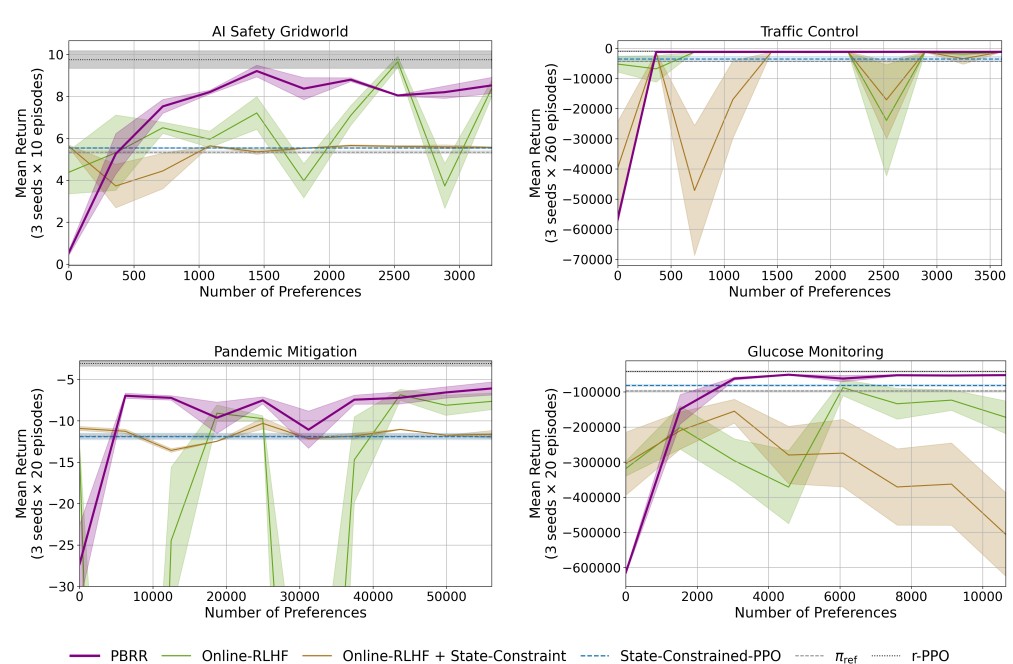

Figure 11: Complementing Figure 2: Mean return under the ground-truth reward function achieved by PBRR compared to baselines that learn a reward function *ab initio* from preferences, averaged over trajectories sampled from policies trained with the learned reward function across 3 random seeds. Shaded regions indicate the standard error. No scaling or clipping is applied to the plotted mean return values.

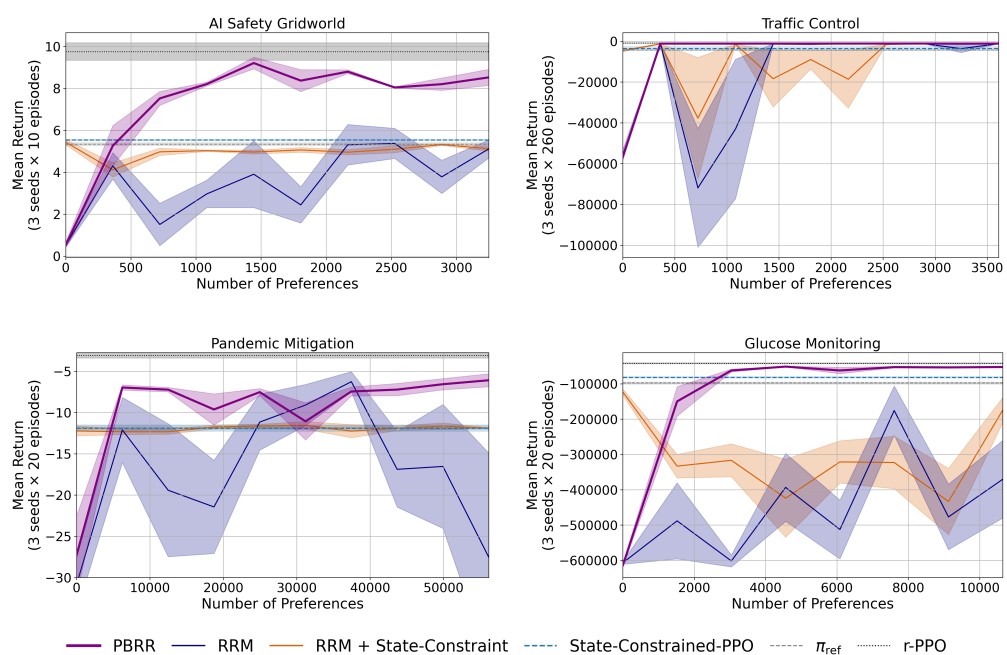

Figure 12: Complementing Figure 2: Mean return under the ground-truth reward function achieved by PBRR compared to other approaches that repair the proxy reward function with preferences, averaged over trajectories sampled from policies trained with the updated proxy reward function across 3 random seeds. No scaling or clipping is applied to the plotted mean return values.

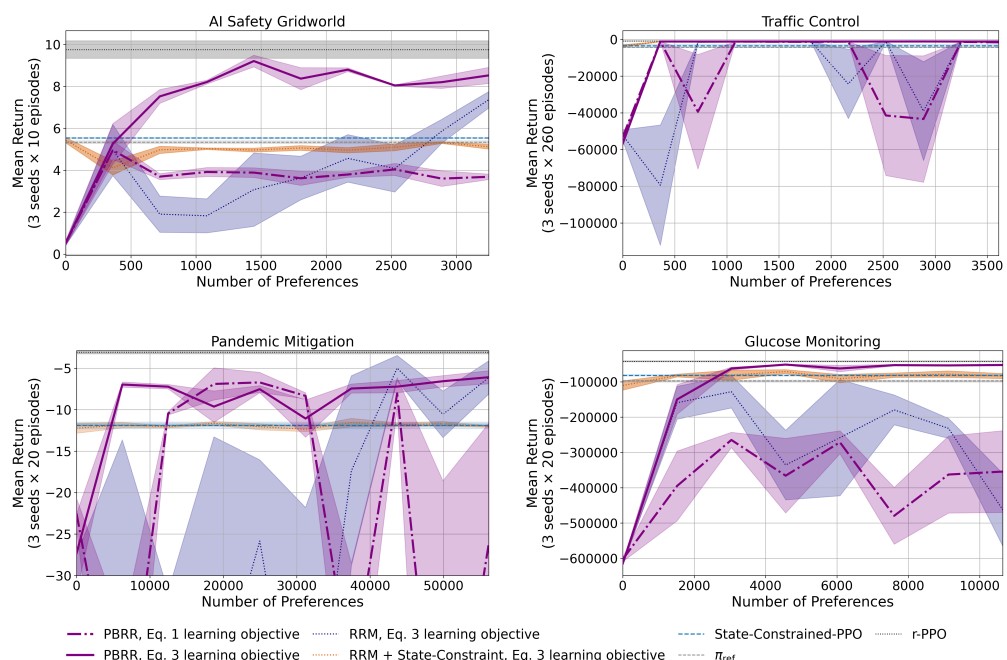

Figure 13: Complementing Figure 3: Mean return under the ground-truth reward function achieved by PBRR, compared to (i) PBRR using the standard preference-learning objective (Eq.1) instead of our proposed objective (Eq.3), and (ii) other methods that repair the proxy reward function equipped with our proposed objective. The mean return is averaged over trajectories sampled from policies trained with the updated proxy reward function across 3 random seeds. No scaling or clipping is applied to the plotted mean return values.

| Method | Initial Mean Return | After Update 1 | After Update 2 |
|---|---|---|---|
| Online-RLHF | $-103.09 \pm 77.24$ | $-141.44 \pm 86.12$ | $-65.27 \pm 61.81$ |
| RRM | $-39.87 \pm 12.82$ | $-9.39 \pm 2.80$ | $-15.37 \pm 6.75$ |
| PBRR | $-26.81 \pm 4.96$ | $-7.91 \pm 1.17$ | $-7.21 \pm 0.84$ |

Table 6: Pandemic Monitoring: mean return over 10 seeds ($\pm$ 95% CI) before and after proxy reward function updates.

better performance than Online-RLHF, and after the second update it achieves statistically significantly better performance than RRM.

## H  AI SAFETY GRIDWORLD QUALITATIVE ANALYSIS

Figure 16 shows the board layout we use for the AI safety gridworld environment. Here we qualitatively analyze why some approaches fail to learn a performant policy in this simple environment. While the other environments are less interpretable than AI safety gridworld, we suspect the methods we consider in this section may share poor performance in those environment for similar reasons.

### H.1  WHY DOES ONLINE-RLHF PERFORM POORLY?

The results in Figure 2 show that the Online-RLHF baseline exhibits oscillatory performance; its performance with respect to the ground-truth reward function decreases upon acquiring new preferences, only to increase again in subsequent iterations. This instability reflects the difficulty of reward learning in these reward-hacking benchmark environments. For instance, after acquiring 1,800 preferences in AI Safety Gridworld, the Online-RLHF baseline observes many trajectory pairs in which visiting more tomato-containing states is always preferred to visiting fewer. As a result, it learns a reward function that assigns positive reward whenever the agent enters a tomato-

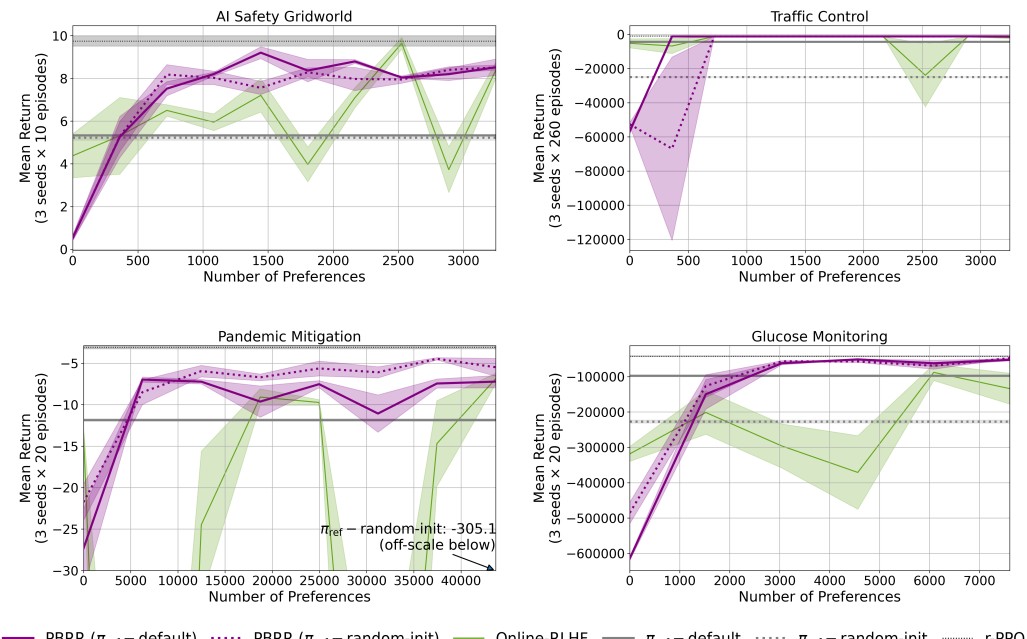

Figure 14: Mean return under the ground-truth reward function achieved by PBRR with the default reference policy used for all other experiments (PBRR ($\pi_{\text{ref}}$ − default)), compared to (i) PBRR using a randomly initialized reference policy (PBRR ($\pi_{\text{ref}}$ − random-init)), and (ii) the Online-RLHF method. The plot for the Pandemic Mitigation environment is truncated to make it easier to differentiate performance between various methods; the full plot is shown in Figure 15.

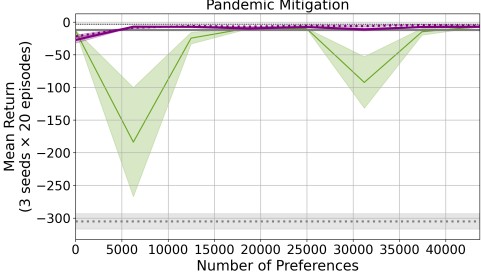

Figure 15: Mean return under the ground-truth reward function achieved by PBRR with the default reference policy used for all other experiments (PBRR ($\pi_{\text{ref}}$ − default)), compared to (i) PBRR using a randomly initialized reference policy (PBRR ($\pi_{\text{ref}}$ − random-init)), and (ii) the Online-RLHF method. For easier visual comparison, we truncate the plot for the Pandemic Mitigation environment in Figure 14.

containing state. In contrast, the ground-truth reward function assigns positive reward only when a tomato is watered—i.e., upon its first visit. Consequently, the learned reward function induces a looping policy that remains in a single tomato-containing state, whereas the ground-truth reward function induces a policy that visits all tomato-containing states. This failure mode illustrates how RLHF methods can conflate instrumental goals (e.g., visiting a tomato-containing state) with terminal goals (e.g., visiting all tomato-containing states) even after learning from a large dataset of preferences; see Marklund et al. (2025) for further characterization of this misalignment type. We suspect that the Online-RLHF baseline learns similarly misaligned reward functions in the other environments, while empirically PBRR does not exhibit this type of misalignment.

## H.2 WHY DOES RRM PERFORM POORLY?

The results in Figure 2 show that the RRM performs poorly in this environment after collecting a substantial number of preferences. This result appears particularly surprising given that Cao et al.

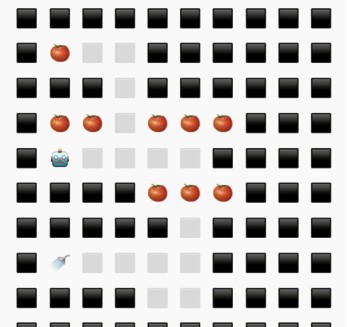

(2025) follow the same methodology and achieve strong results on a suite of robotics tasks. Upon observing the policy that RRM learns after the first iteration, $\pi^*_{\hat{r}_0}$, we note that it only aims to visit the sprinkler state to attain a high reward under the proxy reward function. Trajectories are then sampled from $\pi^*_{\hat{r}_0}$ and used to update the proxy reward function, inducing policy $\pi^*_{\hat{r}_1}$ at the next iteration. Trajectories sampled from $\pi^*_{\hat{r}_0}$ that visit more tomatoes—specifically the ones in coordinates $(5, 4), (5, 5), (5, 6)$ that are on the way to the sprinkler—are always preferred to trajectories that visit less tomatoes. Therefore, the proxy reward function is updated to assign a higher reward for visiting those states. Consequently, the policy derived from the updated proxy reward function does not explore the other states—such as the tomato-containing states in the top half of the grid-world—-because the states including and around $(5, 4), (5, 5), (5, 6)$ are predicted to have higher reward. This process repeats, where the proxy reward function is only ever updated for a particular region of the grid-world—incorrectly assigned high reward—and therefore performance does not improve. More broadly, exploiting the proxy reward function does not necessarily induce an effective exploration policy—an observation also noted by Xie et al. (2024). We suspect Cao et al. (2025) achieve strong performance in their robotics tasks as a result of learning from a proxy reward function that induces a policy that makes meaningful progress toward the true objective. This is notably not the case in the settings we consider, although the proxy reward function can still be updated with relatively few preferences to induce near-optimal performance via PBRR.

### H.3 WHY DOES PBRR WITH THE OBJECTIVE IN EQ. 1 PERFORM POORLY?

The results in Figure 3 show that, in this environment, PBRR with the preference-learning objective in Eq. 1 substantially under-performs PBRR's default implementation, which uses the preference-learning objective in Eq. 3. To understand why, we note that $\pi_{\text{ref}}$ by construction only visits the tomato-containing states at coordinates $(1, 1), (3, 1), (3, 2)$. Throughout training, PBRR observes preferences over trajectories sampled from $\pi_{\text{ref}}$ and $\pi^*_{\hat{r}_t}$. It is usually the case—inevitably during the early iterations—that trajectories from $\pi_{\text{ref}}$ are preferred over trajectories from $\pi^*_{\hat{r}_t}$. Without the regularization terms of Eq. 3, minimizing the cross-entropy loss from Eq. 1 over the collected dataset of preferences then results in high predicted reward being assigned to the transitions encountered by $\pi_{\text{ref}}$. As a result, $\pi^*_{\hat{r}_t}$ begins to visit those transitions at subsequent iterations, but does not explore other parts of the state space such as the tomato-containing states on the right hand side of the grid-world. In effect, minimizing the standard preference loss induces unbounded updates to the proxy reward function, resulting in a policy that over-values the actions taken by $\pi_{\text{ref}}$ and does not explore other, potentially optimal actions. The regularization terms we add for Eq. 3 encourage the updated proxy reward function to remain sufficiently optimistic by encouraging only decrements to the predicted reward when the proxy reward function induces a ranking that doesn't match the elicited preference.

## I ILLUSTRATIVE SCENARIOS: MINIMAL MDPS

In this section we present simple MDPs that highlight PBRR's strengths and weakness respectively.

We first show that there exists MDPs where PBRR can learn an optimal policy much faster than random sampling:

**Theorem I.1.** *There exists an MDP in which PBRR recovers an optimal policy after $O(1)$ preference query, whereas a uniform-exploration baseline requires $O(|S|)$ preferences in expectation.*

*Proof.* **MDP and rewards.** Consider a horizon-$H = 1$ MDP with start state $s_0$. From $s_0$ there are $n$ actions $a_1, \ldots, a_n$ leading deterministically to terminal states $s_1, \ldots, s_n$, respectively. Rewards are tabular over next-states: taking $a_i$ yields immediate reward $r(s_i)$.

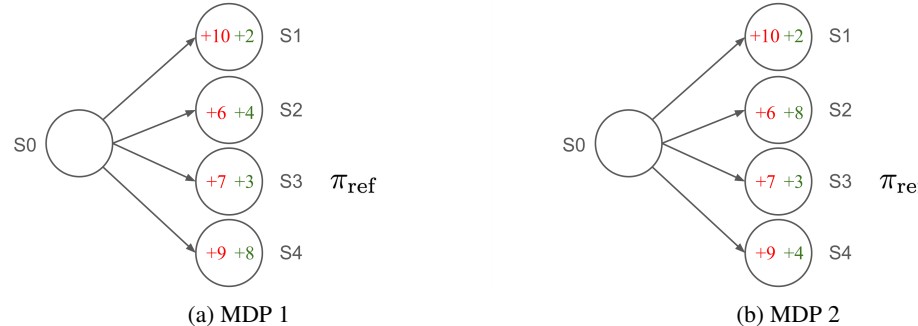

(a) MDP 1 (b) MDP 2

Figure 17: Example MDPs highlighting the strengths and weaknesses of PBRR, as outlined in Appendix I.1. For both MDPs, assume deterministic transition dynamics and 4 available actions from state $s_0$. $\pi_{\text{ref}}$ marks the state that is visited by the supplied reference policy. The proxy reward function's outputted rewards are in red, and the ground-truth reward function's outputted rewards are in green. Rewards are only defined over states and the time horizon $H = 1$.

Let the initial *proxy* reward function satisfy

$$\hat{r}_0(s_1) > \hat{r}_0(s_n) > \hat{r}_0(s_i) \quad \text{for all } i \notin \{1, n\}.$$

Let the *ground-truth* reward function satisfy

$$r(s_n) > r(s_2) > r(s_i) \quad \text{for all } i \notin \{2, n\},$$

so $a_n$ is uniquely optimal. Assume a reference policy $\pi_{\text{ref}}$ is supplied that always chooses action $a_2$. Note that the reference action $a_2$ is strictly better than $a_1$ under $r$. Assume noiseless preferences labeled by $r$ over single-step transitions: $(s_0, a_i, s_i) \succ (s_0, a_k, s_k)$ iff $r(s_i) > r(s_k)$.

**PBRR needs one preference.** First, $\pi^*_{\hat{r}_0}$ is constructed from the initialy proxy reward function $\hat{r}_0$. $\pi^*_{\hat{r}_0}$ chooses $a_1$ (since $\hat{r}_0(s_1)$ is maximal), while the supplied reference $\pi_{\text{ref}}$ deterministically chooses $a_2$. PBRR constructs the pair

$$(s_0, a_1, s_1) \text{ vs. } (s_0, a_2, s_2),$$

which is labeled $(s_0, a_2, s_2) \succ (s_0, a_1, s_1)$ because $r(s_2) > r(s_1)$ by construction. In accordance with Eq. 3, the proxy reward function is updated by decrementing $\hat{r}(s_1)$ below $\hat{r}(s_2)$ while leaving all uncompared transitions unchanged. This update produces $\hat{r}_1$ with

$$\hat{r}_1(s_2) > \hat{r}_1(s_1) \quad \text{and} \quad \hat{r}_1(s_i) = \hat{r}_0(s_i) \text{ for all } i \notin \{1, 2\}.$$

Since initially $\hat{r}_0(s_n) > \hat{r}_0(s_i)$ for all $i \notin \{1, n\}$, and the update leaves $\hat{r}(s_n)$ and $\hat{r}(s_2)$ unchanged, we still have

$$\hat{r}_1(s_n) > \hat{r}_1(s_i) \quad \text{for all } i \neq n.$$

Hence $\pi^*_{\hat{r}_1}$ selects $a_n$, which is optimal under $r$. Therefore PBRR reaches an $r$-optimal policy after a single preference, i.e., $O(1)$.

**Uniform exploration needs $O(|S|)$ preferences.** Consider a baseline that, at each iteration, selects two actions $a_i$ and $a_k$ uniformly from $\{a_1, \dots, a_n\}$ and elicits a preference comparing $(s_0, a_i, s_i)$ against $(s_0, a_k, s_k)$. Assuming the same update rule as PBRR above, the only comparison that can demote the action chosen by the policy induced by the proxy reward function $a_1$ below $a_i$ for $i \neq 1$ (and thereby expose $a_n$ as the new argmax, given the update) is the pair involving $i = 1$. Each iteration hits $i = 1$ with probability $2/n$, so the waiting time to the first such informative comparison is geometric with mean $n/2$. Thus the baseline requires $\mathbb{E}[\#\text{preferences}] = n/2 = O(n) = O(|S|)$ (since $|S| = n + 1$) before it can recover the optimal action $a_n$. $\square$

MDP 1 shows a specific illustration of one such MDP that satisfied Theorem I.1. We now step through the procedure executed by PBRR following Algorithm 1 in MDP 1. First the proxy reward function initially induces a policy that visits $s_1$. Upon observing a preference between this trajectory and the trajectory sampled from $\pi_{\text{ref}}$— $s_3 \succ s_1$— the proxy reward function is updated so that the proxy reward for $s_1$ is lower than the proxy reward for $s_3$. Therefore, the updated proxy reward function at the next iteration induces a policy that visits $s_4$, which is optimal with respect to the ground-truth

reward function. MDP 1 illustrates a scenario where the proxy reward function induces a substantially sub-optimal policy, but updating the proxy reward function at only a single state can repair it so as to induce an optimal policy under the ground-truth reward function. PBRR succeeds in such scenarios.

We next present a scenario, shown in MDP 2 (Figure 17, right) where PBRR with $C = 0$ (no explicit additional exploration) can fail to learn the optimal policy. Here after the same initial step as in MDP, the proxy reward function is updated so that the induced policy visits $s_4$. The preference collected at this iteration will be $s_4 \succ s_3$. Because the objective in Eq. 3 discourages updating the proxy reward function when it induces a ranking that matches the elicited preference, the proxy reward function is not updated further. But the proxy reward function does not induce an optimal policy with respect to the ground-truth reward function. MDP 2 highlights an example where PBRR with $C = 0$ (no explicit additional exploration) will induce a policy that outperforms the reference policy but would not induce optimal performance.

### I.1 THE BENEFITS OF PBRR'S LEARNING OBJECTIVE

Here we present another minimal MDP to illustrate the purpose of the regularization terms we propose in Eq. 3.

**The benefit of $\mathcal{L}^+$**   To illustrate why our objective includes the $\mathcal{L}^+$ regularization term, take the MDP in Figure 18. Assume that we observe the preference $s_3 \succ s_4$, and then update the proxy reward function with the standard cross entropy loss in Eq. 1 rather than our proposed objective. Minimizing this loss will push $\hat{r}(s_3) \to \infty$ and $\hat{r}(s_4) \to -\infty$ as the loss will continue to decrease as $\hat{r}(s_3) - \hat{r}(s_4)$ increases. This may result in an update to the proxy reward function where $\hat{r}(s_3) > \hat{r}(s_1)$, which would result in a policy that goes to state $s_3$ instead of $s_1$. Therefore, even though the proxy reward function correctly ranked states $(s_3, s_4)$ and produced an optimal policy under $r$ before being updated, minimizing the loss in Eq. 1 given the preference $s_3 \succ s_4$ can update the proxy reward function such that it no longer induces an optimal policy under $r$. To avoid this undesirable scenario, we add the regularization term $\mathcal{L}^+$ to discourage updates to the proxy reward function when it induces a correct ranking over trajectories in a pair.

**The benefit of $\mathcal{L}^-$**   Assume that we observe the preference $s_2 \succ s_3$, and then update the proxy reward function with the standard cross entropy loss in Eq. 1 rather than our proposed objective. Minimizing this loss will push $\hat{r}(s_2) \to \infty$ and $\hat{r}(s_3) \to -\infty$ as the loss will continue to decrease as $\hat{r}(s_2) - \hat{r}(s_3)$ increases. This may result in an update to the proxy reward function where $\hat{r}(s_2) > \hat{r}(s_1)$, which would result in a policy that goes to state $s_2$ instead of $s_1$. Therefore, even though the proxy reward function is updated to produce a correct ranking over the pair $(s_2, s_3)$, it no longer induces an optimal policy under $r$. To avoid this scenario, we add the regularization term $\mathcal{L}^-$ to encourage only decrementing the proxy reward function's output (e.g., decreasing $\hat{r}(s_3)$) rather than also increasing its output (e.g., increasing $\hat{r}(s_2)$).

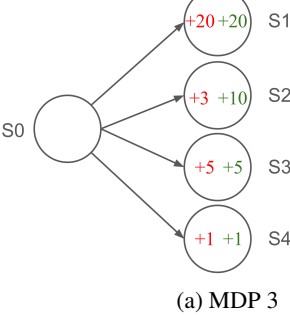

(a) MDP 3

Figure 18: Example MDPs highlighting the benefits of the regluarization terms proposed in Eq. 3, as outlined in Appendix I.1. Assume deterministic transition dynamics and 4 available actions from state $s_0$. The proxy reward function's outputted rewards are in red, and the ground-truth reward function's outputted rewards are in green. Rewards are only defined over states and the time horizon $H = 1$.

## J  REGRET BOUND PROOFS

We now define some additional notation, following Pacchiano et al. (2023). Note when the dynamics model is known, one can compute the expected features $\phi$ under a policy $\pi_i$, which we denote as $\phi(\pi_i)$

$$V_t = \sum_{l=1}^{t-1} (\phi(\tau_l^1) - \phi(\tau_l^2))(\phi(\tau_l^1) - \phi(\tau_l^2))^T + \kappa\lambda I_d \tag{7}$$

$$g_t(w) = \sum_{l=1}^{t-1} \sigma(<\phi(\tau_l^1) - \phi(\tau_l^2), w>)(\phi(\tau_l^1) - \phi(\tau_l^2)) + \lambda W \tag{8}$$

$$w_t^L = \arg\min_{w\,s.t.\,||w||\leq W} ||g_t(w) - g_t(\hat{w}_t^{MLE})||_{V_t^{-1}} \tag{9}$$

$$f_{mk}(\pi_1, \pi_2) = ||\phi(\pi_1) - \phi(\pi_2)||_{V_t^{-1}} \tag{10}$$

$$\alpha_{d,T}(\delta) = 20BW\sqrt{d\log\left(\frac{T(1+2T)}{\delta}\right)} \tag{11}$$

$$\beta_t(\delta) = \sqrt{\lambda}W + \sqrt{\log(1/\delta) + 2d\log\left(1 + \frac{tB}{\kappa\lambda d}\right)} \tag{12}$$

$$C_t(\delta) = w\,s.t.\,||w - w_t^L||_{V_t} \leq 2\kappa\beta_t(\delta) \tag{13}$$

$$\gamma_t(\delta) = 2\kappa\beta_t(\delta) + \alpha_{d,T}(\delta) \tag{14}$$

$$\Pi_t = \left\{\pi_i | (\phi(\pi_i) - \phi(\pi))^T w_t + \gamma_t(\delta)||\phi(\pi^i) - \phi(\pi)||_{V_t^{-1}} \geq 0\ \forall\ \pi\right\} \tag{15}$$

$$\tag{16}$$

When cross entropy loss is used to fit preference data and the reward model is linear, the resulting loss can be expressed as

$$\mathcal{L}_t^\lambda(w) = \sum_{l=1}^t (o_l \log(\sigma(\langle\theta(\tau_l^1) - \theta(\tau_l^2)), w\rangle)) - \frac{\lambda}{2}||w||_2^2 + (1-o_l)\log(\sigma(1 - \langle\theta(\tau_l^1) - \theta(\tau_l^2), w\rangle)), \tag{17}$$

where $o_l = 1$ or $0$ depending on which trajectory is preferred.

As the maximum likelihood estimator $w_{MLE}$ may not satisfy the required boundness assumption, prior work Pacchiano et al. (2023) defined a projected version $w_t^L$ of the weight vector $w$.

We also define the event that the true $w^*$ iles in the specified confidence interval $C_t(\delta)$ on all time steps as

$$\mathcal{E}_\delta = \{\forall t \geq 1, \mathbf{w}_\star \in \mathcal{C}_t(\delta)\}. \tag{18}$$

While in the main text we show the big-O version to avoid the additional notation complexity required to define the terms, we now state a more precise version of Theorem 5.1:

**Theorem J.1.** *Let $\delta \leq 1/$ and $\lambda \geq B/\kappa$. Then under Assumptions 1 and 2, for $f = f_{mk}$ and $\Pi_t = \Pi_{t,mk}$, with probability at least $1 - \delta$, the expected regret of Algorithm 1 is bounded by*

$$Regret_t \leq C_1(4\kappa\beta_t(\delta) + 2\alpha_{d,T}(\delta))\sqrt{2Td\log\left(1 + \frac{TB}{\kappa d}\right)} \tag{19}$$

*Proof.* Our proof closely follows the proof of Theorem 1 in Pacchiano et al. (2023). While their proof was for a different algorithm, we note that their Lemma 7, Corollary 1, Lemma 2 and Lemma 8 all continue to hold in our setting, as they do not depend on the specific policies chosen for exploration.

The first part of the proof of our theorem exactly follows the proof of Theorem 1Pacchiano et al. (2023), where conditioned on the event $\mathcal{E}_\delta$ holding, they bound the regret due to executing the two

exploration policies $\pi_1$ and $\pi_2$:

$$
\begin{aligned}
2r_t &= (\phi(\pi^*) - \phi(\pi_t^1))^\top \mathbf{w}^* + (\phi(\pi^*) - \phi(\pi_t^2))^\top \mathbf{w}^* \\
&= (\phi(\pi^*) - \phi(\pi_t^1))^\top \mathbf{w}_t^L + (\phi(\pi^*) - \phi(\pi_t^1))^\top (\mathbf{w}^* - \mathbf{w}_t^L) + (\phi(\pi^*) - \phi(\pi_t^2))^\top (\mathbf{w}^* - \mathbf{w}_t^L) + (\phi(\pi^*) - \phi(\pi_t^2))^\top \mathbf{w}_t^L \\
&\leq (\phi(\pi^*) - \phi(\pi_t^1))^\top \mathbf{w}_t^L + (\phi(\pi^*) - \phi(\pi_t^2))^\top \mathbf{w}_t^L \\
&\quad + \|\mathbf{w}^* - \mathbf{w}_t^L\|_{V_t} \cdot \|\phi(\pi^*) - \phi(\pi_t^1)\|_{V_t^{-1}} + \|\mathbf{w}^* - \mathbf{w}_t^L\|_{V_t} \cdot \|\phi(\pi^*) - \phi(\pi_t^2)\|_{V_t^{-1}}
\end{aligned}
\tag{20}
$$

They then note that in this sum, the last two terms can be bounded by their Corollary 1:

$$
\|\mathbf{w}^* - \mathbf{w}_t^L\|_{V_t} \cdot \|\phi(\pi^*) - \phi(\pi_t^1)\|_{V_t^{-1}} + \|\mathbf{w}^* - \mathbf{w}_t^L\|_{V_t} \cdot \|\phi(\pi^*) - \phi(\pi_t^2)\|_{V_t^{-1}}
\tag{21}
$$

$$
\leq (2\kappa\beta_t(\delta) + \alpha_{T,d}(\delta)) \cdot \left( \|\phi(\pi^*) - \phi(\pi_t^1)\|_{V_t^{-1}} + \|\phi(\pi^*) - \phi(\pi_t^2)\|_{V_t^{-1}} \right)
\tag{22}
$$

Up to this point, the proof is identical. We now seek to bound the first two terms in Equation 20. First note that

$$
(\phi(\pi^*) - \phi(\pi_t^1))^\top \mathbf{w}_t^L + (\phi(\pi^*) - \phi(\pi_t^2))^\top \mathbf{w}_t^L \leq (2\kappa\beta_t(\delta) + \alpha_{T,d}(\delta)) \left( \|\phi(\pi^*) - \phi(\pi_t^1)\|_{\bar{V}_t^{-1}} + \|\phi(\pi^*) - \phi(\pi_t^2)\|_{\bar{V}_t^{-1}} \right)
\tag{23}
$$

since Line 9 in Algorithm 1 ensures that the selected $\pi_1$ and $\pi_2$ always lie in $\Pi_t$, and therefore holds from the definition of $\Pi_t$.

In addition, Line 9 in Algorithm 1 ensures if $\pi_{\hat{r}_t}^* = \pi_1$ and $\pi_{\text{ref}} = \pi_2$ are used as the exploration policies, then they must satisfy:

$$
\max_{\pi_i, \pi_j \in \Pi_t} f(\pi_i, \pi_j) \leq C_1 f(\pi_{\hat{r}_t}^*, \pi_{\text{ref}}).
\tag{24}
$$

Therefore

$$
\|\phi(\pi^\star) - \phi(\pi_{\text{ref}})\|_{\bar{V}_t}^{-1} + \|\phi(\pi^\star) - \phi(\pi_{\text{proxy}})\|_{\bar{V}_t}^{-1} \leq 2C_1 \cdot \|\phi(\pi_1) - \phi(\pi_2)\|_{\bar{V}_t}^{-1}.
\tag{25}
$$

The remaining part of the proof follows Theorem 1 Pacchiano et al. (2023). Specifically substituting Equations 22, 23 and 25 into Equation 20, and using that $\pi^* \in \Pi_t$ under the assumed event and Lemma 2, yields

$$
2r_t \leq 2(2\kappa\beta_t(\delta) + \alpha_{T,d}(\delta)) \cdot \left( \|\phi(\pi^*) - \phi(\pi_t^1)\|_{V_t^{-1}} + \|\phi(\pi^*) - \phi(\pi_t^2)\|_{V_t^{-1}} \right)
\tag{26}
$$

$$
\leq 4C_1(2\kappa\beta_t(\delta) + \alpha_{T,d}(\delta)) \cdot \|\phi(\pi_t^1) - \phi(\pi_t^2)\|_{V_t^{-1}}
\tag{27}
$$

The remaining few steps in the proof of Theorem 1 then bound

$$
\begin{aligned}
Regret_T &= \sum_t r_t \tag{28} \\
&\leq \sum_t 4C_1(2\kappa\beta_t(\delta) + \alpha_{T,d}(\delta)) \cdot \sum_{t=1}^{T} \|\phi(\pi_t^1) - \phi(\pi_t^2)\|_{V_t^{-1}} \tag{29} \\
&\leq 4C_1(2\kappa\beta_t(\delta) + \alpha_{T,d}(\delta)) \cdot \sqrt{T \sum_{t=1}^{T} \|\|\phi(\pi_t^1) - \phi(\pi_t^2)\|\|_{V_t^{-1}}^2} \tag{30} \\
&\leq 4C_1(2\kappa\beta_t(\delta) + \alpha_{T,d}(\delta)) \cdot \sqrt{2Td \log(1 + \frac{TB}{d})}, \tag{31}
\end{aligned}
$$

using Cauchy-Schwarz for the second inequality and Lemma 8 Pacchiano et al. (2023) for the final inequality. $\qquad\square$

We now provide an analogous proof for the case when the dynamics model is not known. We need some additional notation. Let $N_t(s, a)$ represent the number of times the trajectories have included $(s, a)$ tuples. The proof again follows prior work Pacchiano et al. (2023). They define an alternate covariance matrix that leverages the empirical covariance matrix, and an alternate bonus term and confidence sets needed to account for the uncertainty since the dynamics model is estimated from finite samples.

$$\tilde{\mathbf{V}}_t = \kappa\lambda I_d + \sum_{\ell=1}^{t-1} \left( \phi^{\hat{\mathbb{P}}_\ell}(\pi_\ell^1) - \phi^{\hat{\mathbb{P}}_\ell}(\pi_\ell^2) \right) \left( \phi^{\hat{\mathbb{P}}_\ell}(\pi_\ell^1) - \phi^{\hat{\mathbb{P}}_\ell}(\pi_\ell^2) \right)^\top \tag{32}$$

$$\xi_{s,a}^{(t)}(\eta, \delta) = \min\left( 2\eta, \ 4\eta\sqrt{\frac{U}{N_t(s,a)}} \right), \tag{33}$$

where

$$U = H\log\left(|\mathcal{S}||\mathcal{A}|\right) + \log\left( \frac{6\log(N_t(s,a))}{\delta} \right). \tag{34}$$

The bonus function is:

$$\hat{B}_t(\pi, \eta, \delta) \;\; = \;\; \mathbb{E}_{s_1 \sim \rho, \, \tau \sim \hat{\mathbb{P}}_t^\pi(\cdot|s_1)} \left[ \sum_{h=1}^{H-1} \xi_{s_h,a_h}^{(t)}(\eta, \delta) \right] \tag{35}$$

and the undominated policy set is:

$$\Pi_{t,\mu} = \left\{ \pi^i \; \middle| \; \left( \phi^{\hat{\mathbb{P}}_t}(\pi^i) - \phi^{\hat{\mathbb{P}}_t}(\pi) \right)^\top \mathbf{w}_t^L + \gamma_t \left\| \phi^{\hat{\mathbb{P}}_t}(\pi^i) - \phi^{\hat{\mathbb{P}}_t}(\pi) \right\|_{\tilde{\mathbf{V}}_{t-1}} \right. \tag{36}$$

$$\left. + \hat{B}_t\left( \pi^i, 2SB, \tfrac{\delta}{2|\mathcal{A}||\mathcal{S}|} \right) + \hat{B}_t\left( \pi, 2SB, \tfrac{\delta}{2|\mathcal{A}||\mathcal{S}|} \right) \geq 0, \; \forall\pi \right\}. \tag{37}$$

and we define $f$ as:

$$f_u(\pi_t^1, \pi_t^2) = \gamma_t \left\| \phi_t^{\hat{\mathbb{P}}_t}(\pi^1) - \phi_t^{\hat{\mathbb{P}}_t}(\pi^2) \right\|_{\tilde{\mathbf{V}}_{t-1}} + 2\hat{B}_t(\pi^1, 2WB, \delta) + 2\hat{B}_t(\pi^2, 2WB, \delta). \tag{38}$$

We now restate our Theorem 5.2:

**Theorem J.2.** *(Theorem 5.2) Under Assumptions 5.1,5.3 and 5.4, for $f = f_u$ and $\Pi_t = \Pi_{t,u}$, the regret of Algorithm 1 is bounded by*

$$R_T \leq \tilde{\mathcal{O}}\Big( C_1 \left( \kappa d\sqrt{T} + H^{3/2}\sqrt{|\mathcal{S}||\mathcal{A}|dTH} + H|\mathcal{S}|\sqrt{|\mathcal{A}|dTH} \right) \Big), \tag{39}$$

*Proof.* (sketch) The proof follows the proof of Lemma 15 Pacchiano et al. (2023) with the analogous modification as the one we made in our proof of Theorem 1. Specifically, note that

$$\left\| \phi^{\hat{\mathbb{P}}_t}(\pi^*) - \phi^{\hat{\mathbb{P}}_t}(\pi_t^1) \right\|_{\tilde{V}_t^{-1}} + \left\| \phi^{\hat{\mathbb{P}}_t}(\pi^*) - \phi^{\hat{\mathbb{P}}_t}(\pi_t^2) \right\|_{\tilde{V}_t^{-1}} \leq \max_{\pi_i,\pi_j \in \Pi_{t,u}} \left\| \phi^{\hat{\mathbb{P}}_t}(\pi_i) - \phi^{\hat{\mathbb{P}}_t}(\pi_j) \right\|_{\tilde{V}_t^{-1}} \tag{40}$$

and from Line 9, the definition of $f_u$ ensures:

$$\max_{\pi_i,\pi_j \in \Pi_{t,u}} \left\| \phi^{\hat{\mathbb{P}}_t}(\pi_i) - \phi^{\hat{\mathbb{P}}_t}(\pi_j) \right\|_{\tilde{V}_t^{-1}} \leq C_1 \left( \left\| \phi^{\hat{\mathbb{P}}_t}(\pi^*) - \phi^{\hat{\mathbb{P}}_t}(\pi_t^1) \right\|_{\tilde{V}_t^{-1}} + \left\| \phi^{\hat{\mathbb{P}}_t}(\pi^*) - \phi^{\hat{\mathbb{P}}_t}(\pi_t^2) \right\|_{\tilde{V}_t^{-1}} \right) \tag{41}$$

The remainder of the proof of follows the rest of the proof of Lemma 15 Pacchiano et al. (2023). □

## K  REWARD HACKING ANALYSIS PROOFS

Here we provide a theoretical analysis motivating PBRR, relying on different assumptions than the regret analysis in Section 5.

We show that PBRR's repaired reward function is guaranteed to induce an optimal policy that matches or exceeds the performance of the reference policy (Thm. K.1) and resolve two specific instantiations of reward hacking (Cors. K.2, K.3) in the infinite data setting as the number of iterations goes to infinity.

**Assumption K.1.** *The regularization weights vanish $\lambda_1, \lambda_2 \to 0$ as $t \to \infty$ in Algorithm 1.*

**Assumption K.2.** *Preferences over trajectories are noiseless and determined by the difference in regret between trajectories (Eq. 42).*

We adopt Assumption K.2, following Knox et al. (2022), who show that regret better reflects human preferences—see Appendix K.1 for details.

**Assumption K.3.** $C_1 = 0$ *in Algorithm 1.*

**Assumption K.4.** *Each trajectory set contains the (potentially infinite) support of the corresponding policy, i.e.,* $\text{support}(\pi) \subseteq \mathcal{T}_\pi$.

**Assumption K.5.** *All trajectories begin in the same start state* $s_0$.

**Theorem K.1.** *Suppose assumptions K.2 through K.5 hold, then the optimal policy for the repaired reward function* $\pi_{\hat{r}}^*$ *matches or outperforms the reference policy in the limit as* $t \to \infty$:

$$J_r(\pi_{\hat{r}}^*) \geq J_r(\pi_{ref}).$$

Theorem K.1 implies that reward hacking is resolved as $t \to \infty$, following specific instantiations of the two different definitions of reward hacking from Skalse et al. (2022) and Laidlaw et al. (2025):

**Corollary K.2.** *(No Skalse et al. (2022) Hacking) The reward function* $\hat{r}$ *and the ground-truth reward function* $r$ *are unhackable relative to the optimal policy set for* $\hat{r}$ *and the reference policy* $\pi_{ref}$.

**Corollary K.3.** *(No Laidlaw et al. (2025) Hacking) The reward function* $\hat{r}$ *is unhackable with respect to* $\pi_{ref}$.

## K.1 ASSUMPTIONS

For this theoretical analysis we assume Algorithm 1 is executed with $C_1 = 0$, like we execute in practice.

We define regret under $\tilde{r}$ for a trajectory $\tau = (s_0, a_0, \ldots, s_H)$ as $\text{regret}(\tau \mid \tilde{r}) \triangleq \sum_{t=0}^{|\tau|-1} \gamma^t [V_{\tilde{r}}^*(s_t) - Q_{\tilde{r}}^*(s_t, a_t)] = \sum_{t=0}^{|\tau|-1} (-\gamma^t A_{\tilde{r}}^*(s_t, a_t))$ where $A_{\tilde{r}}^*(s, a) \triangleq Q_{\tilde{r}}^*(s, a) - V_{\tilde{r}}^*(s)$ and $V_{\tilde{r}}^*, Q_{\tilde{r}}^*$ are the optimal value and action-value functions under $\tilde{r}$. We then assume a preference between $\tau_1$ and $\tau_2$ is determined by regret:

$$\mu \triangleq \begin{cases} 0 & \text{if } \text{regret}(\tau_1|r) < \text{regret}(\tau_2|r), \\ 1 & \text{if } \text{regret}(\tau_1|r) > \text{regret}(\tau_2|r), \\ \frac{1}{2} & \text{if } \text{regret}(\tau_1|r) = \text{regret}(\tau_2|r), \end{cases} \tag{42}$$

where regret is the sum of negative optimal advantage:

$$\text{regret}(\tau \mid \tilde{r}) \triangleq \sum_{t=0}^{|\tau|-1} -A_{\tilde{r}}^*(s_t^\tau, a_t^\tau).$$

We also assume $\mathcal{L}_{\text{pref}}$ uses regret-based preferences.

Knox et al. (2022) shows that human preferences are better described by the difference in regret between trajectory segments, as opposed to the difference in the sum of rewards. In practice, we simulate human preference labels using the difference in sum of rewards between trajectories because regret is intractable to compute in our empirical environments, as discussed in Appendix A.

Our assumption about how preferences are labeled (Eq. 42) for this analysis differs in two key ways from Knox et al. (2022): (i) we assume preferences are over trajectories, not trajectory segments (ii) we assume preferences are noiselessly determined by regret, not sampled from a Boltzmann distribution. Our analysis holds without assumption (i) as long as preferences are elicited over an exhaustive set of trajectory segments. We argue that (ii) is a reasonable assumption when executing PBRR, where preferences are only collected between trajectories sampled from $\pi_{\hat{r}_t}$—which is initially substantially sub-optimal due to a misspecified proxy reward function $\hat{r}_t$, and $\pi_{\text{ref}}$—which is assumed to be a safe policy. There is clear distinction between the trajectories from these policies—both qualitatively and in terms of regret under $r$, so we assume no preference noise via Eq. 42.

We also assume that all trajectories begin from the same start state $s_0$. This assumption is without loss of generality: for any MDP with initial state distribution $p_0$, we can construct an equivalent MDP by introducing a new start state $s_0'$ that transitions in a single step according to $p_0$. In this construction, all trajectories then originate from $s_0'$, satisfying the assumption of a single start state.

## K.2 PROOF OF THEOREM K.1

**Lemma K.4.** *Suppose Assumptions K.1 and K.2 hold. Then the repaired reward function $\hat{r}$ and the ground-truth reward function $r$ induce identical regret-based orderings on all inter-policy trajectory pairs $(\tau_1, \tau_2) \in \mathcal{T}_{\pi_{\hat{r}}^*} \times \mathcal{T}_{\pi_{\text{ref}}}$:*

$$\text{sign}\left[\text{regret}(\tau_1 \mid \hat{r}) - \text{regret}(\tau_2 \mid \hat{r})\right] = \text{sign}\left[\text{regret}(\tau_1 \mid r) - \text{regret}(\tau_2 \mid r)\right].$$

*Proof.* Let $(\tau_1, \tau_2) \in (\tau_1, \tau_2) \in \mathcal{T}_{\pi_{\hat{r}}^*} \times \mathcal{T}_{\pi_{\text{ref}}}$ denote an arbitrary inter-policy trajectory pair. Define the Bayes optimal decision rule for distinguishing preferences as

$$f_{0/1}^*(\tau_1, \tau_2; r) \triangleq \begin{cases} 1 & \text{if } p(1 \mid \tau_1, \tau_2, r) > p(-1 \mid \tau_1, \tau_2, r), \\ 0 & \text{if } p(1 \mid \tau_1, \tau_2, r) = p(-1 \mid \tau_1, \tau_2, r), \\ -1 & \text{if } p(1 \mid \tau_1, \tau_2, r) < p(-1 \mid \tau_1, \tau_2, r), \end{cases}$$

where $p(1 \mid \tau_1, \tau_2, r)$ denotes the probability that $\tau_1$ is preferred to $\tau_2$ under the ground-truth reward $r$. Under Assumption K.2, preferences are noiselessly determined by regret, i.e.

$$p(1 \mid \tau_1, \tau_2, r) = \begin{cases} 1 & \text{if } \text{regret}(\tau_1 \mid r) < \text{regret}(\tau_2 \mid r), \\ \frac{1}{2} & \text{if } \text{regret}(\tau_1 \mid r) = \text{regret}(\tau_2 \mid r), \\ 0 & \text{if } \text{regret}(\tau_1 \mid r) > \text{regret}(\tau_2 \mid r). \end{cases}$$

Now, under Assumption K.1, the PBRR loss (Eq. 3) reduces in the limit to the standard preference cross-entropy loss (Eq. 1):

$$\lim_{t \to \infty} \mathcal{L}(g; \hat{r}_{\text{proxy}}, \mathcal{D}) = \mathcal{L}_{\text{pref}}(\hat{r}_{\text{proxy}} + g; \mathcal{D}).$$

Since cross-entropy is a convex margin loss known to be Bayes consistent (Zhang, 2004; Bartlett et al., 2006), any minimizer $\hat{r}$ of $\mathcal{L}_{\text{pref}}$ must induce a predictor $f_{\mathcal{L}_{\text{pref}}}^*$ whose sign agrees with the Bayes optimal rule:

$$\text{sign}\left[f_{\mathcal{L}_{\text{pref}}}^*(\tau_1, \tau_2)\right] = f_{0/1}^*(\tau_1, \tau_2; r).$$

Substituting the regret-based form of $f_{0/1}^*$, it follows that the ordering induced by $\hat{r}$ must coincide with that induced by $r$. Equivalently,

$$\text{sign}\left[\text{regret}(\tau_1 \mid \hat{r}) - \text{regret}(\tau_2 \mid \hat{r})\right] = \text{sign}\left[\text{regret}(\tau_1 \mid r) - \text{regret}(\tau_2 \mid r)\right].$$

$\square$

**Restatement of Theorem K.1.** Suppose Assumptions K.1 through K.5 hold, then the optimal policy for the repaired reward function $\pi_{\hat{r}}^*$ matches or outperforms the reference policy in the limit as $t \to \infty$:

$$J_r(\pi_{\hat{r}}^*) \geq J_r(\pi_{\text{ref}}).$$

*Proof.* Pick any $\tau_1 \in \mathcal{T}_{\pi_{\hat{r}}^*}$ and $\tau_2 \in \mathcal{T}_{\text{ref}}$. By the optimality of $\pi_{\hat{r}}^*$ under $\hat{r}$:

$$\text{regret}(\tau_1 \mid \hat{r}) = 0 \ \leq \ \text{regret}(\tau_2 \mid \hat{r}).$$

By Lemma K.4, this inequality is preserved under $r$, so

$$\text{regret}(\tau_1 \mid r) \ \leq \ \text{regret}(\tau_2 \mid r).$$

Note that this implies

$$\max_{\tau_1} \text{regret}(\tau_1 \mid r) \ \leq \ \min_{\tau_2} \text{regret}(\tau_2 \mid r).$$

On the left hand side, the maximum is an upper bound to the expectation over $\tau_1 \in \mathcal{T}_{\pi_{\hat{r}}^*}$, and the minimum is a lower bound to the expectation over $\tau_2 \in \mathcal{T}_{\pi_{\text{ref}}}$:

$$\mathbb{E}_{\tau_1 \in \mathcal{T}_{\pi_{\hat{r}}^*}}\left[\text{regret}(\tau_1 \mid r)\right] \leq \max_{\tau_1} \text{regret}(\tau_1 \mid r) \ \leq \ \min_{\tau_2} \text{regret}(\tau_2 \mid r) \leq \mathbb{E}_{\tau_2 \in \mathcal{T}_{\pi_{\text{ref}}}}\left[\text{regret}(\tau_2 \mid r)\right].$$

Therefore, by Assumption K.4,

$$\mathbb{E}_{\tau_1 \sim \pi_{\hat{r}}^*}\left[\text{regret}(\tau_1 \mid r)\right] \ \leq \ \mathbb{E}_{\tau_2 \sim \pi_{\text{ref}}}\left[\text{regret}(\tau_2 \mid r)\right].$$

Under the single-start-state assumption (Assumption K.5),

$$\mathbb{E}_{\tau \sim \pi}\big[\text{regret}(\tau \mid r)\big] \;=\; V_r^*(s_0) - V_r^\pi(s_0).$$

Substituting this identity and rearranging gives

$$V_r^*(s_0) - V_r^{\pi_{\hat{r}}^*}(s_0) \;\leq\; V_r^*(s_0) - V_r^{\pi_{\text{ref}}}(s_0).$$

Rearranging terms again gives

$$V_r^{\pi_{\hat{r}}^*}(s_0) \;\geq\; V_r^{\pi_{\text{ref}}}(s_0)$$

which is equivalent to

$$J_r(\pi_{\hat{r}}^*) \;\geq\; J_r(\pi_{\text{ref}}).$$

$\square$

### K.3 THEOREM K.1, COROLLARY K.2

Skalse et al. (2022) defines reward hacking as:

**Definition K.5.** *(Skalse et al., 2022) A pair of reward functions $(\tilde{r}_1, \tilde{r}_2)$ is hackable relative to a policy set $\Pi$ and an environment $(S, A, T, \gamma, p_0, \_)$ if there exists $\pi, \pi' \in \Pi$ such that*

$$J_{\tilde{r}_1}(\pi) > J_{\tilde{r}_1}(\pi') \text{ and } J_{\tilde{r}_2}(\pi) < J_{\tilde{r}_2}(\pi')$$

This canonical definition of reward hacking intuitively states that, if $(\tilde{r}_1, \tilde{r}_2)$ is not hackable given a set of policies $\Pi$, then there does not exist any pair of policies in $\Pi$ such that one policy has a strictly higher expected return under $\tilde{r}_1$ while the other has a strictly higher expected return under $\tilde{r}_2$. Framed differently, any increase in expected return under $\tilde{r}_1$ from switching policies in $\Pi$ never decreases the expected return under $\tilde{r}_2$. This definition is particularly strict, as noted by Skalse et al. (2022).

The proof for Corollary K.2 immediately follows from Theorem K.1 that if $\mathcal{K}(\hat{r}, r; \mathcal{T}_{\pi_{\hat{r}}^*} \cup \mathcal{T}_{\pi_{\text{ref}}}) = 1$ and we define the set of policies as

$$\Pi \;=\; \{\pi_{\text{ref}}\} \cup \Pi_{\hat{r}}^*,$$

where $\Pi_{\hat{r}}^*$ denotes the set of all policies optimal for $\hat{r}$, then the pair $(\hat{r}, r)$ is not hackable relative to $\Pi$ under Definition K.5.

### K.4 THEOREM K.1, COROLLARY K.3

Laidlaw et al. (2024) define reward hacking as:

**Definition K.6.** *(Laidlaw et al., 2024) Suppose $\tilde{r}$ is a $\rho$-correlated proxy[5] with respect to $\pi_{ref}$. The proxy reward function is hackable with respect to the ground-truth reward function and $\pi_{ref}$ if,*

$$J_r(\pi_{\hat{r}}^*) < J_r(\pi_{ref}).$$

This definition of reward hacking only considers reward functions that are reasonable optimization targets, i.e., by requiring the inputted reward function to be $\rho$-correlated with the ground-truth reward function, and sets a threshold for reward hacking via the expected return of the reference policy under the ground-truth reward function. This definition is less strict than that of Skalse et al. (2022); see Laidlaw et al. (2024) for an in-depth comparison,

The proof for Corollary K.3 immediately follows from Theorem K.1 that if $\mathcal{K}(\hat{r}, r; \mathcal{T}_{\pi_{\hat{r}}^*} \cup \mathcal{T}_{\pi_{\text{ref}}}) = 1$, then $\hat{r}$ is not hackable with respect to $r$ and $\pi_{\text{ref}}$ under Definition K.6.

---

[5]For the full definition of $\rho$-correlation, we refer readers to the original paper from Laidlaw et al. (2024).

