# OpenReview forum: "Repairing Reward Functions with Human Feedback to Mitigate Reward Hacking"
_ICLR.cc/2026/Conference — Submitted to ICLR 2026_

### Official Review · Reviewer_azDK · 2025-10-16

**Soundness:** 2
**Presentation:** 3
**Contribution:** 2
**Rating:** 4
**Confidence:** 4

**Summary:**

This paper proposes a method, PBRR, that adjusts a human-provided proxy reward function by learning an additive correction term. This additive term is learned from human preferences using a novel preference learning objective (not the typical cross entropy one). The authors demonstrate in reward hacking benchmarks that PBRR can perform similarly and might outperform strict preference learning methods or other methods that modify a proxy reward.

**Strengths:**

The paper tackles a relevant and important problem: improving human-provided proxy reward functions to make them better aligned with true human objectives. Reward design is very difficult; therefore, developing methods to improve human designed reward functions is useful.

The proposed loss function is conceptually simple (that's a good thing!) and the decomposition into three loss terms provides interpretability about how the correction is applied.

The authors provide both theoretical and empirical support to their approach.

**Weaknesses:**

**Assumptions about human-provided rewards**:
The paper assumes that humans can readily specify proxy reward functions that reflect the ground-truth objective but lack robustness. However, existing empirical evidence (e.g., Booth et al., 2023; Muslimani et al., 2025) suggests that humans often provide misaligned or poorly specified reward functions. The assumption that a proxy reward is “close to optimal” appears unjustified.

**Strong modeling assumptions**:
The method assumes that the ground-truth reward can be expressed as the sum of the proxy reward and a correction term. I'm not sure how realistic this assumption is, especially in high-dimensional or partially observable tasks where misspecification can be complex and non-additive.

**Loss function**:
The paper introduces three loss terms, but the behavior of the second and third terms is unclear.

The second loss term, $L_{+}$, is intended to regularize the correction term $g$ towards zero on trajectory pairs where the proxy reward function agrees with the human preference. However, from the definition provided, this loss would also be minimized when
$g(\tau_1) = -g(\tau_2)$,
which could alter the overall reward function and potentially invert or misclassify previously correct preferences.

The third loss term, $L_{-}$, explicitly prioritizes decreasing the reward for undesirable behaviors rather than increasing the reward for desirable behaviors. The motivation for this asymmetric treatment is unclear. The authors should discuss what effect this design choice has on the learned correction $g$ and the resulting policy behavior.

**Assumptions about the reference policy**:
The method assumes access to a “good enough” reference policy that can distinguish failures of the proxy reward-induced policy. This also seems like a strong assumption. This means that the reference policy can’t make the same mistakes that the proxy induced policy makes. Moreover, in the regret analysis, the authors further assume that the reference policy lies in the set of possibly optimal policies induced by the ground truth reward function. Lastly, the appendix mentions that some reference policies were obtained from “only a handful” of demonstrations. It is unclear how many demonstrations this entails.

**Experimental design and baselines:**
- The choice of reward learning baselines is limited. The paper omits more recent or preference-efficient algorithms such as PEBBLE (Lee et al., 2021) or recent RLHF approaches, which would provide stronger comparisons.The RLHF baseline provided in the text (Christiano et al. (2017)) is notoriously preference inefficient.

- The comparison between PBRR and Online-RLHF seems unfair: PBRR benefits from a hand-designed proxy reward (which itself requires human effort), while Online-RLHF learns from scratch, so it makes sense that it would require more human preferences.

- The normalization strategy for the figures seems unconventional and lacks justification.
- Statistical significance is unclear: results are based on only three seeds, and shaded regions represent standard error rather than 95% confidence intervals. Therefore, claims that PBRR “matches or exceeds” all baselines are not statistically supported.

**References**:

Booth et al (2023): https://scholar.google.com/citations?view_op=view_citation&hl=en&user=sf3ROEUAAAAJ&citation_for_view=sf3ROEUAAAAJ:KlAtU1dfN6UC

Muslimani et al (2025): https://arxiv.org/abs/2503.05996

Lee et al (2021): https://arxiv.org/pdf/2106.05091

Christiano et al. (2017): https://arxiv.org/abs/1706.03741

**Questions:**

- What empirical evidence supports the assumption that humans can easily specify a proxy reward that reflects the true objective (albeit imperfectly)?
- Can the authors clarify how the $L_{+}$ loss avoids pathological cases where $g(\tau_1) = -g(\tau_2)$?
- Why did the authors decide to focus on correcting undesirable transitions  (that have a high reward) rather than correcting desirable ones (that may incorrectly have a low reward)?
- In the online RLHF baseline, the authors note that they use the reference policy for initial exploration. What does that mean? Do the authors initialize the starting policy with the reference policy?
- Could the authors justify the normalization strategy used in their plots?
- Are the reported improvements statistically significant?
- Why were stronger baselines such as PEBBLE not included?

---

> ### Author Response · Authors · 2025-11-21
>
> Thank you for your in-depth review!.
>
> ***Assumptions about human-provided rewards being “close to optimal”:*** We apologize this was unclear and we completely agree that it would be unjustified to assume a human provided reward function is “close to optimal”. In all environments we evaluate with, the proxy reward function induces significantly sub-optimal performance; in fact, it is often worse than a uniformly random policy. To clarify, PBRR assumes the proxy reward function reflects the ground-truth objective in that it correctly assigns reward to at least some transition, but not that it induces good performance. Per your comment, in Section 6.1 we have added additional clarification.
>
> ***Strong modeling assumptions:*** As you state, we do assume the ground-truth reward function can be represented as the sum of the proxy reward function and a correction term. We hope to correct your claim, however, that this is limiting; this modeling assumption is without loss of generality and can represent any Markov reward function. To see why, let $r(s,a,s’)$ define a ground-truth reward function. We can equivalently represent this ground truth as the sum of $\hat{r}(s,a,s’) + g(s,a,s’)$ where $\hat{r}(s,a,s’) = 0$ and $g(s,a,s’) = r(s,a,s’)$.
>
> ***Loss function terms explanations:*** We appreciate your in-depth look at our proposed objective, and agree that the role of each term in our proposed loss function could be better explained! We have added clarifications about the role of each regularization term, including an intuitive example to highlight the undesirable scenarios these terms prevent, in Appendix I.1.
>
> To highlight the importance of each term in our proposed objective, we note that Figure 3 illustrates that without our proposed regularization terms, PBRR’s performance is significantly degraded. In Appendix G.4 we also show that PBRR performs best when using both regularization terms rather than just one.
>
> ***Loss function pathological case:*** The pathological case you describe where the loss is potentially minimized when $g(\tau_1) = -g(\tau_2)$ is fortunately not a concern because the loss term aims to minimize the sum of squares $g(\tau_1)^2 + g(\tau_1)^2$, which can only be accomplished by pushing $g(\tau_1)$ and $g(\tau_2)$ to 0.
>
> ***Assumptions about the reference policy:*** We apologize that we were not clear – we assume access to a reference policy, but it is not essential that this is a “good” or performant reference policy. To show this empirically, in Appendix G.8, we added an additional ablation showing that PBRR – when using a randomly initialized policy as the reference policy– continues to match or outperform other baselines. These results indicate that a randomly initialized reference policy is “good enough” because, as you point out, it doesn’t “make the same mistakes that the proxy induced policy” does. We have now also clarified this in Section 6.3.
>
> Our regret analysis does assume the reference policy lies in the set of possibly optimal policies induced by the ground truth reward function—a limitation of our analysis that is not realized in practice. Algorithm 1 handles this assumption by falling back to the policy pair constructed on Algorithm 1 Line 9 when the assumption is not met.
>
> ***Reference policy missing training detail:*** We have updated our appendix to specify exactly how many demonstrations were used to train the reference policy. Thanks for catching this missing detail!
>
> ***The choice of reward learning baselines is limited:*** Thanks for the suggestion! In Appendix F we have added extensive additional ablations of the Online-RLHF baseline, including PEBBLE and two other more recent approaches. We show that our method is still more preference-efficient, and generally that these data-efficiency methods—which were designed for robotics domains—struggle in the reward hacking benchmark we consider.
>
> ***The comparison between PBRR and Online-RLHF seems unfair:*** A key takeaway from our work is that we can benefit from leveraging a misspecified proxy reward function when learning from preferences, which prior work such as Online-RLHF has not considered. You wrote that “it makes sense that [online-RLHF] would require more human preferences [than benefitting from a hand-designed proxy reward]”. While one might assume this is the case, actually methods that correct a proxy reward do not always outperform online RLHF. In particular, we show in Figure 2 that some other methods for repairing a proxy reward function (e.g., the RRM  (Cao et al. 2025) and RRM + State-Constraint baselines)  do not necessarily do better than Online-RLHF. In contrast, our PBRR method matches or improves upon performance of Online-RLHF and these and other baselines, and is more data efficient.  We have updated our description of the RRM baseline in Section 6.2 to make the connection to Online-RLHF more clear.

---

> ### Author Response · Authors · 2025-11-21
>
> ***Normalization strategy justification:*** We use the normalization strategy from Knox et al., 2022 to make our main figures easier to parse; for all plots, a y-axis value of 0 indicates the performance of the supplied reference policy and a value of 1 indicates the performance of an optimal policy with respect to the true reward. Figures 9, 10, and 11 plot the same results from the paper without the normalization strategy applied; the same patterns are evident.
>
> ***Statistical significance:*** We are in full agreement that establishing statistical significance would strengthen our work and we are in the process of doing so. Given the rebuttal period timeline, our desire to respond to concerns by the ICLR suggested Nov 20 timeline, and our available computational resources, we have prioritized first running results for Pandemic Monitoring . In Appendix G.9, we have added initial results for the Pandemic environment where we compute 95% confidence intervals, demonstrating PBRR’s performance relative to the best-performing baseline: after the first reward function update, PBRR produces a reward function that yields statistically significantly better performance than Online-RLHF, and after the second update it achieves statistically significantly better performance than RRM.
>
> We will include this analysis for all domains in the camera-ready version. We generally expect that PBRR will continue to match or outperform all baselines when running with more seeds.
>
> ***Additional questions:***
> >In the online RLHF baseline, the authors note that they use the reference policy for initial exploration. What does that mean? Do the authors initialize the starting policy with the reference policy?
>
> Thanks for highlighting that this was not clear. We both initialize the starting policy with the reference policy and initialize the Online-RLHF’s candidate batch to also contain trajectories sampled from the supplied reference policy. Additional details are in Appendix D.2.
>
> >Why did the authors decide to focus on correcting undesirable transitions (that have a high reward) rather than correcting desirable ones (that may incorrectly have a low reward)?
>
> We begin with the hypothesis that humans are more likely to specify reward functions that incorrectly assign high reward (i.e., optimistic reward functions). For example, a human-specified reward function where the human has forgotten to add some penalty—such as a robot using excessive gripping force or an LLM responding with too many emojis—is an optimistic proxy reward function.
>
> Note that while PBRR is conceptually motivated by an optimistic proxy, it does not fundamentally require this condition, and none of our theoretical results rely on it. In the Glucose Monitoring environment, the proxy reward function we evaluate with (Figure 2) breaks our optimism assumption but PBRR still outperforms all other baselines in this environment, and in Appendix F.6 we empirically show that PBRR still outperforms the other baselines when the optimism assumption is not met in the AI Safety Gridworld

---

### Official Review · Reviewer_w4tN · 2025-10-31

**Soundness:** 3
**Presentation:** 4
**Contribution:** 3
**Rating:** 6
**Confidence:** 4

**Summary:**

This paper proposes Preference-Based Reward Repair (PBRR), a method to mitigate reward hacking in reinforcement learning by iteratively correcting a human-specified proxy reward function instead of learning a reward model from scratch. PBRR learns an additive correction term over transitions based on human trajectory-pair preferences and uses a targeted exploration strategy comparing the learned policy and a reference policy. The method introduces a tailored objective that focuses corrections on transitions where the proxy reward conflicts with preferences. The authors provide theoretical cumulative regret bounds in tabular MDPs comparable to uncertainty-based RLHF methods and empirically evaluate PBRR on reward-hacking benchmarks. Results show improved preference efficiency and policy performance over standard RLHF and proxy-modification baselines.

**Strengths:**

Strengths:
- Reward hacking and costly preference collection are established issues. The framing of “repairing” instead of replacing reward functions is interesting
- The proposed loss explicitly distinguishes between aligned and misaligned transitions, encouraging targeted corrections rather than global reward learning.
- Empirical results demonstrate that PBRR requires fewer human preferences to achieve high-performing policies than RLHF baselines.
- Regret bounds: Theoretical analysis establishes sublinear cumulative regret comparable to prior strategic preference-based RL methods.
- Method leverages existing proxy reward functions and can use imperfect reference policies which is more realistic.

**Weaknesses:**

Weakness:
- Though the formulation of "repairing" is interesting, the paper can benefit from a more thorough discussion between this paradigm and algorithms that learn a human reward in addition to the original (sparse) environment reward. Some references that can help illustrate this:
	- Zhang et al., GUIDE: Real-Time Human-Shaped Agents (continuous feedback shaping and simulator-based human reward modeling combined with existing environment reward)
	- Peng et al., Learning from Active Human Involvement through Proxy Value Propagation (active intervention and human-guided policy shaping, alternative to preference-based repair)
- The method presumes the proxy reward is “aligned or overly optimistic”, which may not hold universally. The effects of pessimistic or adversarial proxies are underexplored or need a discussion.
- Experiments mainly use reward-hacking benchmarks; lack of evaluation on tasks with continuous control or high-dimensional observations. Could the authors discuss more on this and how the proposed method can be generalized to these domains? Many references mentioned in the related works have already conducted experiments in these domains.
- The algorithm relies on a reference policy for exploration guidance, but quality requirements and failure cases are not evaluated.
- Preferences are assumed to follow a Bradley–Terry model; discussions on the robustness to noisy or inconsistent human feedback should be better addressed.

**Questions:**

Addressing some of the weakness would be very helpful.

---

> ### Author Response · Authors · 2025-11-21
>
> We thank you for your review..
>
> **More thorough discussion between this paradigm and algorithms that learn a human reward:** Thank you for the good suggestion. We have added the papers you suggested to our related works section, as well as an additional discussion on related methods that make restrictive assumptions about what human feedback is available:
>
> >Second, other methods assume access more demanding human feedback, such as corrective actions \citep{jiang2024reinforcement,peng2023learning}, continuous-valued human ratings \citep{zhang2024guide}, or feature-attribution-based explanations \citep{mahmud2023explanation}. Relying on corrective actions would require the human reward designer to provide demonstrations---e.g., controlling a fleet of autonomous vehicles on a highway or determining appropriate pandemic lockdown policies---which demands substantial expertise. Continuous-valued feedback and explanation-based supervision similarly impose a high cognitive burden, limiting who can design aligned reward functions. Our approach, by contrast, requires the human to provide comparative judgments.
>
> In the above peng2023learning and zhang2024guide are the two papers you suggested
>
> **Optimistic reward function:**
> We completely agree that humans may not always provide optimistic rewards, and we apologize for not being clear that, while our method may be particularly beneficial when the provided reward function is accurate or optimistic in most states, that
>
> - Our algorithm does not require the reward function to be optimistic;
>
> - Both our sets of theoretical results (Thm 5.1 and 5.2,  Thm J.1) do not assume the input reward is optimistic;
>
> - Our experimental results show our PBRR method still does well in domains where the input reward is not aligned or optimistic. Specifically, in the Glucose Monitoring environment, our input proxy reward function is not optimistic but PBRR still outperforms all other baselines in this environment (Figure 2). Similarly, in Appendix F.6 we show our PBRR still outperforms the other baselines when the optimism assumption is not met in the AI Safety Gridworld.
>
> We have updated the paper to clarify (a) and (b) in Section 4, and (c) in Section 6.3. Our empirical results therefore demonstrate that the optimism assumption is not essential to the performance gains achieved by PBRR.
>
>
> ***Lack of evaluation with continuous control/high-dim state space:*** We apologize this was unclear – actually these reward hacking environments include continuous action (Glucose Monitoring, Traffic Control) and high-dimensional state spaces (Glucose Monitoring, Traffic Control, Pandemic Mitigation). Table 1 provides a summary of the state and action spaces for each environment. To recap Table 1:
>
> - Pandemic Mitigation: 312-dim continuous state; discrete action ∈ {−1, 0, 1}
>
> - Glucose Monitoring: 96-dim continuous state; 1-dim continuous action ∈ [0, 1]
>
> - Traffic Control: 50-dim continuous state; 10-dim continuous action ∈ [0, 1]
>
> - AI Safety Gridworld: 36-dim discrete state; discrete action ∈ {0, 1, 2, 3}
>
> For comparison, the state spaces for the Glucose Monitoring, Traffic Control, and Pandemic Mitigation environments are more than 2.4x, 8x, and 1.28x larger, respectively, than that of the popular Meta-World robotics environments. The action space for Traffic-Control is 2.5x larger. We have added this clarification to the paper in Section 6.1
>
> **Evaluating the reference policy quality and impact on results:** Thank you for the helpful suggestion. In Appendix G.8, we added an additional ablation showing PBRR’s performance when using a randomly initialized policy as the reference policy. With a randomly initialized reference policy, PBRR continues to match or outperform other baselines. These results show that a randomly initialized policy provides sufficient exploration guidance and therefore can be used as a reference policy. We have now also clarified this in Section 6.3.
>
> **Discussion on limitations of Bradley-Terry preferences:** In Section 2, we have added a brief disclaimer about our assumption that preferences follow the Bradley-Terry preference model and cited a paper that details further critiques:
>
> >Although ubiquitous, this model of noisy rationality may not account for all the ways in which humans fail to act optimally; see \citet{zhi2025beyond} for further discussion.

---

### Official Review · Reviewer_P69g · 2025-11-01

**Soundness:** 3
**Presentation:** 3
**Contribution:** 3
**Rating:** 8
**Confidence:** 2

**Summary:**

This paper addresses the problem of reward hacking in RL. The paper introduces Preference-Based Reward Repair (PBRR), an iterative framework that aims to "repair" an initial, imperfect proxy reward function. PBRR works by learning an additive, transition-dependent correction term.

The method has two core component: (1) a targeted exploration strategy that elicits human preferences on trajectory pairs generated by the current (repaired) policy; (2) a new preference-learning objective that consists of three terms: $\mathcal{L}_{pref}$ (the standard cross-entropy preference loss), $\mathcal{L}^+$ (a regularization term that penalizes corrections for trajectories where the proxy reward already agrees with the human preference, and $\mathcal{L}^-$ (a regularization term that, when the proxy disagrees with the preference, penalizes corrections on the preferred trajectory)

The authors provide a theoretical regret analysis in tabular domains, showing a variant of PBRR matches sublinear regret bounds of prior work. Empirically, PBRR is shown to be significantly more data-efficient and stable, outperforming baselines that learn from scratch (RLHF) or use alternative repair methods (like the concurrent Residual Reward Modeling, RRM). The authors also thoroughly ablate their results

**Strengths:**

- The two key components of PBRR are novel and well-justified. The optimism-based loss function (Eq. 3) is a clever piece of mechanism design, directly targeting the assumed cause of reward hacking (over-optimism) by prioritizing negative corrections
- The paper presents good empirical results and thorough ablations
- The paper is well-written and presents good additional information in the Appendix, eg in App. G, explaining why strong baselines like RRM and Online-RLHF fail

**Weaknesses:**

- The empirical evaluation is somewhat limited in complexity. The environments are useful but relatively simple. The paper would be significantly strengthened by demonstrating PBRR's effectiveness on more complex, high-dimensional continuous control domains
- The main regret analysis in Section 5 is for a complex algorithm variant (with C_1 > 0) that is not used in the experiments (which set C_1 = 0)
- The method relies on $\pi_{ref}$. The paper uses reasonable BC-derived policies. However, the sensitivity to the quality of $\pi_{ref}$ is not explored. What if $\pi_{ref}$ is a random policy? An ablation on the quality of $\pi_{ref}$ would strengthen the paper's claims

**Questions:**

See Weaknesses

---

> ### Author Response · Authors · 2025-11-21
>
> Thank you for your review and encouraging remarks about our paper!
>
> ***Empirical evaluation is somewhat limited in complexity:*** Thank you for this feedback. We selected these 4 domains as they constitute a challenging  reward hacking benchmark used by important work in this field (Laidlaw et al., 2025, Pan et al., 2022), including continuous action (Glucose Monitoring, Traffic Control) and high-dimensional state spaces (Glucose Monitoring, Traffic Control, Pandemic Mitigation). Table 1 provides a summary of the state and action spaces for each environment. For comparison, the state spaces for the Glucose Monitoring, Traffic Control, and Pandemic Mitigation environments are more than 2.4x, 8x, and 1.28x larger, respectively, than that of the popular Meta-World robotics environments. The action space for Traffic-Control is 2.5x larger. To recap Table 1:
>
> - Pandemic Mitigation: 312-dim continuous state; discrete action ∈ {−1, 0, 1}
> - Glucose Monitoring: 96-dim continuous state; 1-dim continuous action ∈ [0, 1]
> - Traffic Control: 50-dim continuous state; 10-dim continuous action ∈ [0, 1]
> - AI Safety Gridworld: 36-dim discrete state; discrete action ∈ {0, 1, 2, 3}
>
> We have added this clarification to the paper in Section 6.1.
>
> ***Sensitivity of PBRR to the performance of the reference policy/ ablation on how its quality impacts results:*** Thank you for the interesting suggestion.  Per your suggestion, in Appendix G.8, we now show that a randomly initialized policy often suffices as a useful reference policy; PBRR continues to match or outperform all baselines when using a randomly initialized reference policy. We have now also clarified this in Section 6.3.

---

### Official Review · Reviewer_9csQ · 2025-11-03

**Soundness:** 2
**Presentation:** 3
**Contribution:** 2
**Rating:** 2
**Confidence:** 5

**Summary:**

The work presents a method for correcting a misspecified proxy reward function in the broad framework of PBRL and RLHF automatically by designing a new loss function. The work’s proposed PBRR (preference based reward repair) is claimed to be better than RLHF as it requires less data than full RLHF, and easily incorporate a human specified, potentially misaligned reward. Empirically, the work demonstrates their method can learn better than learning a reward function from scratch using preference data using fewer data samples, and archives high performing policies.

**Strengths:**

The work is well motivated. I also liked the idea of humans specifying an initial, potentially misaligned rewards. Then using targeted exploration, the methods can automatically correct the reward function to get true, hidden human prerefences.

**Weaknesses:**

There are some concerns in the problem formulation, solution design and empirical setup that are noted below.

One main issue in section 4 is that the authors make an assumption in lines 182-190 that humans provide an overly optimistic reward function. This insight is certainly not true. While in some cases humans can be overly optimistic which may cover some past cases of reward hacking in Krakovna et al, understanding how humans misspecify the reward is itself a monumental challenge. Some humans can be more risk sensitive, some more risk seeker. Thus, framing the entire learning objective, which is the main contribution of the work, in Eq 3 on the overly optimistic scenario is not general, and can be misleading in a range of real world domains. Thus, this is one major issue with the  optimistic reward hypothesis with the propose formulation.

Unfortunately, the para “Constructing a preference dataset” is described only in passing, and several important details missed. This step, how to construct a pref dataset to repair the proxy reward is highly important and what is the novel contribution of the work over prior cited work needs to be specified clearly, and proper ablations must be done over different methods to acquire data to correct proxy reward function.

Experimentally, the work is tested only on 4 domains, which is quite limited evaluation. Furthermore, despite the main claim of the paper as humans specifying the proxy reward function \hat{r}, in the tested domains the proxy reward seems to be hardcoded using the logic in table 1. This is not inspiring confidence that the approach can really learn from real world human feedback as humans are often not good at providing numerical score to their preferences. Thus, empirical evaluation currently is not strong.

Furthermore, experiments on all the real human labeled data in the context on RLHF for LLMs is required. It also solves the problem of getting real human rewards, as human specified preferences are already collected in various publicly available dataset. This dataset can be used to get a proxy reward.

Better justification for when having access to a base policy \pi_ref that specifies safe behavior is possible. In the context of LLMs, it is not clear if such a based policy will exist.

Overall, I find the work addresses an important problem. I find value in the approach. Main outstanding issues remain using strong assumptions on how humans specify rewards to design a solution method, and experiments that are limited and do not use real human feedback, which seems inconsistent with the main premise of the paper.

**Questions:**

See above

---

> ### Author Response · Authors · 2025-11-21
>
> Thank you for your detailed feedback and thoughtful review!
>
> ***Optimistic reward function:***
> We completely agree that humans may not always provide optimistic rewards, and we apologize for not being clear that, while our method may be particularly beneficial when the provided reward function is accurate or optimistic in most states, that
>
> - Our algorithm does not require the reward function to be optimistic;
>
> - Both our sets of theoretical results (Thm 5.1 and 5.2,  Thm J.1) do not assume the input reward is optimistic;
>
> - Our experimental results show our PBRR method still does well in domains where the input reward function is not aligned or optimistic. Specifically, in the Glucose Monitoring environment, our input proxy reward function is not optimistic but PBRR still outperforms all other baselines in this environment (Figure 2). Similarly, in Appendix F.6 we show our PBRR still outperforms the other baselines when the optimism assumption is not met in the AI Safety Gridworld.
>
> We have updated the paper to clarify (a) and (b) in Section 4, and (c) in Section 6.3. Our empirical results therefore demonstrate that the optimism assumption is not essential to the performance gains achieved by PBRR.
>
> ***Details on constructing a preference dataset and ablations over different methods:***
> Thank you for raising this concern about the paragraph on preference dataset construction missing details. Actually we did include all details in Algorithm 1, Lines 5–12, but we have revised the text around Algorithm 1 to make this explicit.
>
> We completely agree ablations are needed to compare across methods. In our originally submitted paper Figure 3 compares performance under the same PBRR objective but with two alternative preference dataset construction methods from prior work (“RRM, Eq.3 learning objective” and “RRM + State Constraint, Eq.3 learning objective”). In the paper, we refer to these as different “exploration strategies,” but we have revised this to be clearer that this means "methods for constructing a preference dataset.” PBRR with our proposed preference dataset construction procedure (specified in Algorithm 1) outperforms both two alternative methods. If there are different ablations for ways to construct preference datasets that you were thinking of, please just let us know.
>
> ***Concerns with evaluation on only 4 domains:*** Thank you for this feedback. We selected these 4 domains as they constitute a challenging  reward hacking benchmark used by important work in this field (Laidlaw et al., 2025, Pan et al., 2022), including continuous action (Glucose Monitoring, Traffic Control) and high-dimensional state spaces (Glucose Monitoring, Traffic Control, Pandemic Mitigation). Table 1 provides a summary of the state and action spaces for each environment. For comparison, the state spaces for the Glucose Monitoring, Traffic Control, and Pandemic Mitigation environments are more than 2.4x, 8x, and 1.28x larger, respectively, than that of the popular Meta-World robotics environments. The action space for Traffic-Control is 2.5x larger. We have added this clarification to the paper in Section 6.1. If there was a particular domain you had in mind beyond these sets, we welcome the suggestion.
>
> ***Experiments with real human data in RLHF for LLMs:*** We completely agree that human experiments would be a great step for future work. We appreciate your suggestion to leverage an existing human labeled dataset for RLHF in LLM tasks, but we wish to clarify that our focus is on sequential multi-turn decision making tasks. LLM preference datasets are on single turn or few turn tasks which may not exhibit the reward-hacking behaviors we study due to their short task-horizon. Note that in our setting, running a human experiment would be extremely demanding. This sort of iterative training with humans is exciting, but is a substantial research effort involving additional infrastructure and challenges that we leave for future work.
>
> ***Justification for access to a “safe behavior” reference policy:*** We apologize that we were not clear – we assume access to a reference policy, but it is not essential that this is a “good” or performant reference policy. We have clarified this in Section 6.3.  To show this empirically, in Appendix G.8, we added an additional ablation showing that PBRR – when using a randomly initialized policy as the reference policy– continues to match or outperform other baselines.

---

> > ### Author Response · Authors · 2025-11-21
> >
> > ***The proxy reward function is hardcoded:*** We weren’t quite clear what the reviewer meant by this, but we did want to make sure to clarify the problem setting we focus on: we are primarily interested in settings where a human can specify a proxy (imperfect) reward function via hardcoded logic. This is the canonical RL setting. As you point out, “humans are not good at providing numerical scores”; this is precisely the problem PBRR addresses by repairing a proxy reward function with preferences. Our method never solicits reward values from humans, but rather assumes humans input a flawed proxy reward function.

---

> > ### Comment · Reviewer_azDK · 2025-11-21
> > **Clarification on the Experiments**
> >
> > Based on Reviewer 9csQ's comments, I realize I may have misunderstood the experiment setup:
> >
> > Are the proxy reward functions that need to be “repaired” the ones provided in the reward hacking benchmark environments?
> >
> > How were the preferences gathered—were these actual human preferences?

---

> ### Author Response · Authors · 2025-11-21
>
> The proxy reward functions that need to be “repaired” are the ones provided in the reward hacking benchmark environments as examples of misspecified reward functions.
>
> The preferences gathered are not actual human preferences; they are synthetically labeled (i.e., sampled from the Boltzmann distribution under the ground-truth reward function, which is a standard model of human preferences). We note that PEBBLE (Lee et al., 2021) and Christiano et al., 2017 follow a similar experimental setup, as does follow-up work in Online-RLHF.

---

> > ### Comment · Reviewer_azDK · 2025-11-21
> >
> > This raises concerns about the validity of the results (in terms of whether this approach can actually work with humans). While past work has often relied on simulated preferences, I strongly believe this is a flawed standard and should not continue. Human preferences are inherently noisy and inconsistent, which makes them substantially different from simulated preferences derived from a ground-truth reward. For example, using a ground-truth reward can produce preferences between trajectories that differ only marginally, but humans are unlikely to reliably distinguish such small differences.

---

> > > ### Author Response · Authors · 2025-11-21
> > >
> > > We definitely agree that running human experiments would strengthen our—and other works—empirical results. However, collecting human preferences is extremely expensive and presents many practical implementation challenges, particularly for online-methods like ours. For example, after each round of preference elicitation, we would need to compute a new reward function, retrain a new policy, generate new trajectories, and then ask the same participant for another round of preference judgements. These practical constraints are well-known, which is why many influential contributions to online RLHF evaluate with synthetic preference labels.
> > >
> > > We believe that we should still make algorithmic contributions to RLHF without requiring prohibitively costly human experiments. Our work proposes a paradigm for repairing reward functions with human feedback, and validates our method empirically with synthetic human feedback—-a standard in this field—while also providing a theoretical analysis to corroborate our empirical results. Running human studies is an exciting next step.

---

> > > > ### Comment · Reviewer_azDK · 2025-11-21
> > > >
> > > > I completely understand that user studies require significant time and effort. However, if gathering human preferences in a way that can be directly used for your algorithm is not feasible, this suggests that, in practice, your method relies on synthetic feedback. Moreover, there are online preference-learning studies that have successfully conducted experiments with humans [1–3]. Overall, if this method is intended to work with real humans, the submission should provide evidence demonstrating its effectiveness in that setting.
> > > >
> > > > [1] Christiano, P. F., Leike, J., Brown, T. B., Martic, M., Legg, S., & Amodei, D. (2017). Deep reinforcement learning from human preferences. Advances in Neural Information Processing Systems.
> > > >
> > > > [2] White, D., Wu, M., Novoseller, E., Lawhern, V. J., Waytowich, N., & Cao, Y. (2024). Rating-Based Reinforcement Learning. Proceedings of the AAAI Conference on Artificial Intelligence.
> > > >
> > > > [3] Muslimani, C., & Taylor, M. E. (2025). Leveraging sub-optimal data for human-in-the-loop reinforcement learning. International Conference on Learning Representations.

---

> > > > > ### Author Response · Authors · 2025-11-24
> > > > >
> > > > > Thank you for continuing to engage on this point. We fully agree that real human preferences are noisy, imperfect and generally different from synthetic preferences (generated over the ground-truth reward function). That said, we’d like to argue that synthetic preferences are a feature, not a bug: it gives a controlled setting to answer our core algorithmic question—can we repair a human-specified proxy reward from preferences in a data-efficient way? Starting with synthetic labels lets us run extensive ablations, including to address your other concerns and to isolate the our proposed learning objective and targeted exploration, without confounding effects from a full human-in-the-loop pipeline.

---

### Author Response · Authors · 2025-12-02

**For the AC:** below we restate the key additions to our original submission and rebuttals to common concerns among reviewers. We address the concerns unique to a single reviewer individually.

**Assumptions about the reference policy:** All reviewers requested additional discussion or ablations regarding the reference policy required by PBRR for exploration. We apologize that we were not clear – we assume access to a reference policy, but it is not essential that this is a “good” or performant reference policy. To show this empirically, in Appendix G.8, we added an additional ablation showing that PBRR – when using a randomly initialized policy as the reference policy– continues to match or outperform other baselines.

**Optimistic reward function:** Reviewer 9csQ and w4tN raised concerns that our method assumes the inputted proxy reward function is optimistic. We completely agree that humans may not always provide optimistic rewards, and we apologize for not being clear that, while our method may be particularly beneficial when the provided reward function is accurate or optimistic in most states, that
Our algorithm does not require the reward function to be optimistic;
Both our sets of theoretical results (Thm 5.1 and 5.2,  Thm J.1) do not assume the input reward is optimistic;
Our experimental results show our PBRR method still does well in domains where the input reward function is not aligned or optimistic. Specifically, in the Glucose Monitoring environment, our input proxy reward function is not optimistic but PBRR still outperforms all other baselines in this environment (Figure 2). Similarly, in Appendix F.6 we show our PBRR still outperforms the other baselines when the optimism assumption is not met in the AI Safety Gridworld.
We have updated the paper to clarify (a) and (b) in Section 4, and (c) in Section 6.3. Our empirical results therefore demonstrate that the optimism assumption is not essential to the performance gains achieved by PBRR.

**Concerns about simplicity of the domains used for evaluation:** Reviewer 9csQ, p69g, and w4tN raised concerns regarding the simplicity of the four domains we use to evaluate our method. We selected these 4 domains as they constitute a challenging reward hacking benchmark used by important work in this field (Laidlaw et al., 2025, Pan et al., 2022), including continuous action (Glucose Monitoring, Traffic Control) and high-dimensional state spaces (Glucose Monitoring, Traffic Control, Pandemic Mitigation). Table 1 provides a summary of the state and action spaces for each environment. For comparison, the state spaces for the Glucose Monitoring, Traffic Control, and Pandemic Mitigation environments are more than 2.4x, 8x, and 1.28x larger, respectively, than that of the popular Meta-World robotics environments. The action space for Traffic-Control is 2.5x larger. We have added this clarification to the paper in Section 6.1.

**Synthetic human preference labels:** Reviewer 9csQ and azDK raised the concern that our empirical results use preference labels sampled from a model of human preferences (i.e., the Boltzmann distribution), but are not real human preference labels. We highlight that it would be prohibitively costly to elicit preferences in the real-world domains we consider (e.g., pandemic lockdown regulation design, insulin administering for type II diabetes), particularly given the online nature of our problem setting. Assuming synthetic preference labels enables us to make important algorithmic contributions to Online-RLHF and run extensive ablations. Our strong theoretical and empirical results motivate future human studies focused on understanding how to repair human specified reward functions, but this direction warrants a separate investigation.

---

### Meta-Review · Area_Chair_p341 · 2026-01-09

**Summary:**

his paper studies reward hacking and proposes a framework for repairing proxy rewards using preferences, with supporting theory and experiments on standard benchmarks. The problem is important and the approach is conceptually clear.

However, several key concerns remain unresolved. The empirical evaluation relies entirely on synthetic preference labels, and the absence of evidence with real human feedback raises fundamental questions about practical validity; this issue was explicitly reiterated in post-rebuttal discussion. Additional concerns include limited evaluation scope, mismatches between theoretical assumptions and experimental settings, and insufficient empirical support for robustness to noisy or inconsistent human feedback. For reviewers who did not engage post-rebuttal, their original concerns remain outstanding.

Reviewer opinions were mixed with marginal scores, and no explicit post-rebuttal score increases were indicated. Due to the interrupted process, this assessment reflects a conservative synthesis based only on the available record.

**Reviewer Concerns:**

Reviewer 9csQ

Addressed: Clarification that the method does not require an optimistic proxy reward (theory + counter-examples).

Outstanding: Lack of real human preference data; limited evaluation scope; strong assumptions on human-specified rewards and reference policy.
(No post-rebuttal confirmation.)

Reviewer P69g

Partially Addressed: Reference policy sensitivity (random policy ablation); clarification of domain complexity.

Outstanding: Theory–experiment mismatch (regret analysis uses C1 > 0; experiments use C1 = 0).
(No post-rebuttal confirmation.)

Reviewer w4tN

Partially Addressed: Related-work discussion; optimism assumption; reference policy dependence.

Outstanding: Robustness to noisy/inconsistent human feedback.
(No post-rebuttal confirmation.)

Reviewer azDK

Partially Addressed: Modeling assumptions, loss behavior, baselines, normalization, limited statistical analysis.

Outstanding: Core validity concern due to reliance on synthetic (non-human) preferences; reviewer explicitly maintained this concern post-rebuttal.

**Reviewer Scores:**

Reviewer 9csQ

Original: 2

Likely post-rebuttal: 2

Rationale: Major realism and human-feedback concerns remain; no post-rebuttal signal.

Reviewer P69g

Original: 8

Likely post-rebuttal: 8

Rationale: No explicit indication of score change.

Reviewer w4tN

Original: 6

Likely post-rebuttal: 6

Rationale: Key robustness concern unresolved; no explicit signal.

Reviewer azDK

Original: 4

Likely post-rebuttal: 2–4

Rationale: Reviewer reiterated objections to synthetic preferences and lack of human validation.

---

### Decision · Program_Chairs · 2026-01-26

Reject